# Zhyper: Factorized Hypernetworks for Conditioned LLM Fine-Tuning

## Abstract

Large Language Model (LLM) conditioning refers to instructing an LLM to generate content in accordance with the norms and values of a specific culture, beliefs of a particular political orientation, or any desired text-specified semantic conditioning. Unfortunately, prompt engineering does not ensure that LLMs behave in accordance with a desired conditioning due to the inductive bias of the pre-training and alignment datasets. Prior works have focused on fine-tuning LLMs by directly conditioning the LoRA weights; however, such methods introduce a large number of parameters. As a remedy, we propose Zhyper, a parameter-efficient factorized **hyper**network framework that generates context-aware LoRA adapters from textual descriptions. Experiments on multiple benchmarks show that Zhyper achieves competitive performance with up to **26x** fewer parameters than the state-of-the-art baselines. Furthermore, we extend Zhyper to cultural alignment, demonstrating improved generalization to out-of-domain settings and a better capturing of fine-grained contextual values.

## 1 Introduction

Large Language Models (LLMs) have transformed Natural Language Processing (NLP), Computer Vision (CV), and machine learning (ML) more broadly. They achieve state-of-the-art performance in text generation and comprehension across diverse domains, including code synthesis (Rozière et al., 2023), mathematical reasoning (Ahn et al., 2024), scientific writing (Geng et al., 2025; Eger et al., 2025), multimodal tasks such as text–image understanding and generation (Alayrac et al., 2022), and evaluation of machine translation and related tasks (Gu et al., 2025). This success stems from scaling to millions and billions of parameters. However, this scaling requires large computational resources, motivating the search for parameter-efficient fine-tuning (PEFT) techniques.

Recent advances have made it possible to adapt LLMs to task-specific criteria, which is crucial for a broader applicability and acceptance of NLP systems. A recent stream of research leverages PEFT techniques (Ding et al., 2023; Weyssow et al., 2023; Prottasha et al., 2024; Wang et al., 2025; Loeschcke et al., 2024; Yang et al., 2024), e.g., Low-Rank Adaptions (LoRA) (Hu et al., 2021) to adapt for desired task-specific values in an LLM. LoRA achieves this by freezing most of the pre-trained model's parameters and introducing trainable low-rank matrices, yielding weight correction terms. However, stand-alone LoRA approaches are primarily tailored for a single-task adaptation and may lose their effectiveness in a setting where an LLM needs to be adapted to various downstream settings. Therefore, approaches directly tackling a multi-task learning (MTL) setting have been proposed (Agiza et al., 2024; Wang et al., 2023; Luo et al., 2024; Wang et al., 2024) that aim to do multi-task fine-tuning efficiently, where a shared backbone model must serve multiple tasks. A promising direction for the dynamic and robust individualization of LLMs is by leveraging *hypernetworks* in the training pipeline. In Text-to-LoRA (T2L) (Charakorn et al., 2025), the authors apply hypernetworks to adapt LLMs to specific task descriptions using only a textual task description as the input for learning the adapters' weights. However, two open challenges remain unresolved. First, existing conditioned LoRA methods, such as T2L, are not parameter-efficient when extended to large contextual spaces. Second, the applicability of conditioned LoRA tuning has not been explored for the important real-world problem of cultural alignment.

To tackle the described challenges, we propose a factorized **hyper**network, called **Zhyper**, which leverages a hypernetwork to inject desired values in the outputs of an LLM. More specifically, the

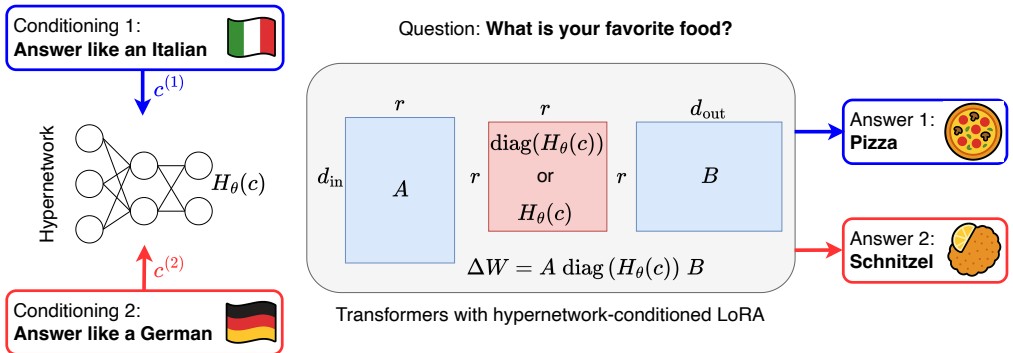

Figure 1: We introduce a novel parameter-efficient architecture for conditioned LLM finetuning based on hypernetwork-conditioned LoRA adapters

hypernetwork should produce a different weight based on the current layer, a layer type for attention-awareness, and the respective description of a context we want to adapt to. As opposed to prior works (Charakorn et al., 2025), we additionally experiment with contexts being descriptions of cultures. Considering the example shown in Figure 1, the goal is to condition a base model on certain criteria. For instance, when choosing a preferred food, the answer might have country-specific dependence. The contextual modulation signal is computed via a hypernetwork that is integrated into the computation of the LoRA adapter, leading to answers conditioned on the instilled values.

We empirically show that our novel model achieves comparable predictive performance at an order of magnitude fewer parameters on a variety of LLM capability assessments, e.g., math, science, coding, reasoning, and word knowledge. Furthermore, we provide a thorough ablation study on the contextual modulation signal represented as an $(r \times r)$-matrix, where $r$ denotes the rank of the LoRA adapters.

Our contributions are as follows:

- A novel lightweight hypernetwork-based architecture for training LoRA adapters that align to text or culture descriptions with up to **26x** fewer parameters compared to prior work.
- Hypernetwork that generates a compact contextual modulation signal instead of generating all parameters of an adapter.
- A thorough empirical study on efficient learning strategies for the conditioned fine-tuning of Large Language Models.
- Improved empirical performances in the important use cases of task conditioning and cultural alignment.

## 2 ZHYPER - CONDITIONED LLM TUNING

Our method Zhyper leverages hypernetworks to induce descriptive information and generate LoRA adapters for context-specific adaptation. The following subsections present the preliminaries in Section 2.1, our novel factorized architecture in Section 2.2, and the complexity analysis of our method in Appendix D.

### 2.1 PRELIMINARIES.

**Low-Rank Adaptation (LoRA)** is a well-established parameter-efficient fine-tuning technique for LLMs (Hu et al., 2021). Generally, the weights of a base model are frozen, and only low-rank weight matrices are trained, serving as adapters to the model. Formally, for each selected linear transformation $h = W^{\text{base}} x$, the fine-tuned transformation is given by $h' = W^{\text{base}} x + \Delta W x$, with $\Delta W = AB$, where $A \in \mathbb{R}^{d_{\text{in}} \times r}$, and $B \in \mathbb{R}^{r \times d_{\text{out}}}$ are low-rank weight matrices with $r \ll d$. VeRA (Kopiczko et al., 2024) modifies this formulation by introducing trainable scaling vectors

$d \in \mathbb{R}^r$ and $b \in \mathbb{R}^{d_{\text{in}}}$, expressed as diagonal matrices $\Lambda_d$ and $\Lambda_b$, while freezing $\boldsymbol{A}$ and $\boldsymbol{B}$. This yields the update $\Delta\boldsymbol{W} = \boldsymbol{\Lambda}_b \boldsymbol{A} \boldsymbol{\Lambda}_d \boldsymbol{B}$. LoRA-XS (Bałazy et al., 2024) instead initializes the low-rank matrices using truncated SVD of the base weight matrix, $\boldsymbol{W}^{\text{base}} = \boldsymbol{U}_r \boldsymbol{\Sigma}_r \boldsymbol{V}_r^T$, setting $\boldsymbol{A} = \boldsymbol{U}_r \boldsymbol{\Sigma}_r$ and $\boldsymbol{B} = \boldsymbol{V}_r^T$ and keeping them frozen. It then trains a square matrix $R \in \mathbb{R}^{r \times r}$, resulting in the update $\Delta\boldsymbol{W} = \boldsymbol{U}_r \boldsymbol{\Sigma}_r \boldsymbol{R} \boldsymbol{V}_r^T$.

**Hypernetworks** introduce neural networks whose output defines the parameters of another network (Ha et al., 2016). Generally, it formalizes the idea of learning to generate weights rather than learning weights directly. Formally, let $f_\psi(\cdot)$ denote a parameterized target network with $\psi \in \mathbb{R}^n$. A parameterized hypernetwork $H_\phi(\boldsymbol{v}): \mathbb{R}^m \to \mathbb{R}^n$ by weights $\phi$ maps an input embedding or context vector $\boldsymbol{v} \in \mathbb{R}^m$ to a set of parameters $\psi$ for a target network.

## 2.2 ARCHITECTURE

We present the **Zhyper** method, a hypernetwork-conditioned low-rank adaptation method that enables parameter-efficient and context-aware fine-tuning of LLMs. The general workflow of our method is illustrated in Figure 1, where in the following we provide details on the respective components.

**Contextual Information.** We represent contextual features (e.g., value or cultural descriptions) leveraging a transformer-based encoder trained for general text embeddings. Each description is transformed into a fixed-length embedding vector $\boldsymbol{c} \in \mathbb{R}^{d_c}$, which serves as the contextual input to our hypernetwork described below. This representation ensures that diverse textual descriptions are mapped into a unified semantic space suitable for conditioning LoRA adapters. We denote by $\boldsymbol{c}_i$ the contextual information associated with the $i$-th dataset.

**Factorized Hypernetworks (Zhyper-diag).** Let $\mathbb{D} = \{D_i\}_{i=1}^n$ be fine-tuning datasets, where $D_i = \{(\boldsymbol{X}_i, \boldsymbol{Y}_i)\}$ is a set of input-label pairs. Each dataset $i$ is associated with a set of contextual descriptions $\mathbb{C}_i := \{\boldsymbol{c}_i^{(j)}\}_{j=1}^M$ where $\boldsymbol{c}_i^{(j)} \in \mathbb{R}^{d_c}$. During training, we sample $D_i \sim \mathbb{D}$ and $\boldsymbol{c}_i \sim \mathbb{C}_i$.

For each transformer layer $\ell \in \{1, \ldots, L\}$ and attention projection $t \in \{Q, V\}$ of the base LLM, we learn module-type and layer-specific embeddings. For that, we utilize learnable embeddings $e_t = E_{type}(t) \in \mathbb{R}^{d_t}$ and $e_\ell = E_{layer}(\ell) \in \mathbb{R}^{d_\ell}$, shared across training. Our hypernetwork $H_\phi^{\text{vec}}: \mathbb{R}^{d_c + d_t + d_\ell} \to \mathbb{R}^r$ is defined to map the concatenated input to a rank-$r$ vector:

$$\boldsymbol{z}_{\ell,t}^i = H_\phi^{\text{vec}}(\boldsymbol{c}_i^{(j)} \| e_t \| e_\ell) \tag{1}$$

where $\|$ denotes the concatenation operator. Intuitively, $\boldsymbol{z}_{\ell,t}^i \in \mathbb{R}^r$ denotes a latent representation of a contextual encoding for the $i$-th dataset w.r.t. the $\ell$-th layer and the attention component $t$, i.e., query or value projections. This leads to the following update rule for the base model's weights:

$$\Delta\boldsymbol{W}_{\ell,t}(c) = \boldsymbol{A}_{\ell,t} \operatorname{diag}(\boldsymbol{z}_{\ell,t}^i) \boldsymbol{B}_{\ell,t} \quad \text{with} \quad \boldsymbol{A}_{\ell,t} \in \mathbb{R}^{d_{\text{in}} \times r}, \boldsymbol{B}_{\ell,t} \in \mathbb{R}^{r \times d_{\text{out}}} \tag{2}$$

$$\boldsymbol{W}_{\ell,t}^{\text{adapt}} \boldsymbol{x} \leftarrow (\boldsymbol{W}_{\ell,t}^{\text{base}} + \Delta\boldsymbol{W}_{\ell,t}) \boldsymbol{x} \tag{3}$$

where $\operatorname{diag}(\boldsymbol{z}_{\ell,t}^i) \in \mathbb{R}^{r \times r}$ yields a diagonal matrix with the elements of $\boldsymbol{z}_{\ell,t}^i$ on the diagonal.

The **Zhyper-square** variant is an ablation of our method where the hypernetwork outputs a square matrix $H_\phi^{\text{sq}}: \mathbb{R}^{d_c + d_t + d_\ell} \to \mathbb{R}^{r \times r}$, leading to $\Delta\boldsymbol{W}_{\ell,t}(c) = \boldsymbol{A}_{\ell,t} \boldsymbol{z}_{\ell,t}^i \boldsymbol{B}_{\ell,t}$ where $\boldsymbol{z}_{\ell,t}^i \in R^{r \times r}$.

**Training Objective.** To integrate the hypernetwork-generated LoRA adapters into the base model with weights $\boldsymbol{W}^{\text{base}}$, we formalize the training objective as minimizing the supervised fine-tuning loss over datasets and their associated contextual descriptors, ensuring that each layer and module type is conditioned on context-specific information. We define the trainable parameters $\theta = \{A_{\ell,t}, B_{\ell,t}, \phi, E_{type}, E_{layer}\}$. The supervised fine-tuning training objective becomes:

$$\arg\min_\theta \mathbb{E}_{i \sim [n]} \mathbb{E}_{(x,y) \sim \mathbb{D}_i} \mathbb{E}_{c_i^{(j)} \sim \mathbb{C}_i} \mathcal{L}_{\text{SFT}}\left( f_{\boldsymbol{W}^{\text{base}}, \Delta\boldsymbol{W}\left(c_i^{(j)}\right)}(x), y \right) \tag{4}$$

where $f_{\boldsymbol{W}^{\text{base}}, \Delta\boldsymbol{W}(c_i^{(j)})}$ denotes our model's output given the frozen weights of the base model $\boldsymbol{W}^{\text{base}}$ and $\Delta\boldsymbol{W}(c_i^{(j)})$ denoting the adaptation according to Equation (2) for the $j$-th contextual descriptor

fo the $i$-th dataset. The architecture of our framework enables training the matrices $\boldsymbol{A}$ and $\boldsymbol{B}$ once, whereas the hypernetwork provides an efficient contextual modulation by either providing a diagonal scaling matrix or a full square matrix. Zhyper can be applied to any variant of LoRA where $\Delta \boldsymbol{W}$ is decomposed into two matrices.

# 3 EXPERIMENTS

In our experimental protocol, we address two important real-world use cases:

- **Task Conditioning**: where LLMs are conditioned to perform a certain task, e.g., to act as an expert on geography, similar to the setting of T2L (Charakorn et al., 2025) (Section 3.1).

- **Cultural Alignment**: where LLMs are instructed to generate content aligned with the norms and values of a culture, e.g., to write like a European (Section 3.2).

• *Hyperparameters of our method.* We use a 3-layer MLP, with the weight of output head of size $d_{\text{MLP\_out}} \times r$ which is different from T2L head, with weight of $d_{\text{MLP\_out}} \times r \times (d_{\text{out}} + d_{\text{in}})$ where $d_{\text{MLP\_out}}$ is the output size of the last MLP block. To generate the embeddings of the text descriptions, we use `gte-large-en-v1.5` (Zhang et al., 2024; Li et al., 2023). Our method introduces a new hyperparameter, $Z$ matrix type, which can be either a diagonal matrix or a square matrix. Using this hyperparameter together with the LoRA rank, we conduct a hyperparameter analysis on a subset of the benchmark dataset (validation set). We find that the configuration with $r = 8$ and a diagonal $Z$ matrix achieves the best performance on 10 task-based benchmark subsets while maintaining a low number of parameters ($\sim 4.2$M). In evaluation, we refer to this variant as simply **Zhyper**. We perform a similar hyperparameter tuning procedure for the cultural alignment models. Comparisons between different variants are provided in Appendix B. All experiments in this section are based on standard LoRA (Hu et al., 2021). A comparison using VeRA (Kopiczko et al., 2024) is provided in Appendix C.

The source code of our framework and experiments is publicly available.[1]

## 3.1 USE CASE ON TASK CONDITIONING

• *Baselines.* We evaluate our method on `Mistral-7B-Instruct-v0.2` (Jiang et al., 2023) as an unconditioned baseline model. Additional experiments on `Llama-3.1-8B-Instruct` (Grattafiori et al., 2024) and `Gemma-2-2B-Instruct` (Team et al., 2024) are provided in Appendix C. We further compare against two enhanced variants of the baseline: one augmented with few-shot in-context learning (ICL) (Brown et al., 2020; Dong et al., 2024), and another that incorporates prepended task descriptions in the query. As fine-tuned models, we compare against T2L (SFT) (Charakorn et al., 2025), which performs instant adaptation of LLMs from task descriptions; multi-task LoRA (MTL), a variant of LoRA trained on all tasks; task-specific LoRA (Oracle), trained only on the corresponding task; and Hyperdecoders (Ivison & Peters, 2022), which generate LoRAs on a per-sequence basis. We also report the zero-shot results of Arrow Routing (Ostapenko et al., 2024); because code is unavailable, we copy their reported numbers, which use LoRA rank $r$ of 4. Our experiments show that the best-performing T2L variant uses $r = 16$, while the best MTL variant uses $r = 8$. For completeness, we also report results for LoRA ranks $r = 8$, $r = 16$, and $r = 32$.

• *Datasets.* We use the SNI dataset (Wang et al., 2022) to **train** our task-based model. Following the T2L setup, 11 tasks are held out for evaluation, and 10 datasets are removed to avoid data contamination with the evaluation benchmarks, leaving 479 datasets for training. We also reuse the task descriptions generated in T2L, with 128 descriptions per training dataset. For **evaluation**, we utilize 10 benchmark datasets that enable a broad assessment across diverse areas, such as reasoning, math, science, coding, and world knowledge. We evaluate on the following benchmarks: Arc-challenge (ArC) and Arc-easy (ArE) (Clark et al., 2018), OpenBookQA (OQA) (Mihaylov et al., 2018), HumanEval (HE) (Chen et al., 2021), HellaSwag (HS) (Zellers et al., 2019), MBPP (Austin et al., 2021), Winogrande (WG) (Sakaguchi et al., 2021), GSM8K (Cobbe et al., 2021),

---

[1] `https://anonymous.4open.science/r/Zhyper-F432`

Table 1: Benchmark performance on unseen tasks and descriptions. T2L, MTL, and Task-specific LoRAs results are reproduced by us, while the others are taken from T2L (Charakorn et al., 2025). All methods use a LoRA rank of $r = 8$, except for Arrow Routing, which uses $r = 4$ and T2L with $r = 16$. Best numbers per column are in **bold**.

| | Trainable Params | ArcC (acc) | ArcE (acc) | BQ (acc) | HS (acc) | OQA (acc) | PIQA (acc) | WG (acc) | MBPP (pass@1) | GSM8K (acc) | HE (pass@1) | Avg. (10 tasks) |
|---|---|---|---|---|---|---|---|---|---|---|---|---|
| **Zero-shot adaptation without fine-tuning** | | | | | | | | | | | | |
| Mistral-7B-Instruct | N/A | 65.4 | 77.8 | 71.6 | 49.7 | 54.2 | 72.8 | 45.0 | 43.1 | 40.9 | 37.2 | 55.8 |
| Prepending task desc. | N/A | 72.0 | 85.8 | 67.6 | 58.9 | 63.4 | 77.9 | 59.0 | 41.6 | 40.9 | 39.0 | 60.6 |
| **Few-shot adaptation without fine-tuning** | | | | | | | | | | | | |
| 3-shot ICL | N/A | 72.1 | 85.9 | 71.7 | 59.0 | 66.2 | 76.2 | 58.0 | 42.6 | 40.9 | 37.2 | 61.0 |
| **Zero-shot adaptation after fine-tuning** | | | | | | | | | | | | |
| Arrow Routing ($r = 4$) | N/A | 60.9 | 86.2 | 87.6 | 80.8 | 48.6 | 83.0 | **68.5** | 50.2 | N/A | 28.7 | N/A |
| Hyperdecoders | 55.0M | 76.6 | 88.5 | 83.9 | 65.2 | 76.6 | 81.3 | 64.9 | 51.6 | 43.6 | 40.9 | **67.3** |
| MTL | 3.4M | 74.0 | 87.3 | 84.0 | 63.4 | 69.2 | 81.5 | 60.5 | 49.1 | 47.5 | 39.6 | 65.4 |
| **Fine-tuned directly on test tasks (Oracle)** | | | | | | | | | | | | |
| Task-specific LoRAs | 3.4M | 74.6 | 88.3 | **88.0** | **87.9** | **77.4** | **86.1** | 57.0 | 47.9 | **50.2** | N/A | N/A |
| **Conditioned zero-shot adaptation after fine-tuning** | | | | | | | | | | | | |
| T2L (SFT) L ($r = 16$) | 110.0M | 74.5 | **87.7** | 85.5 | 64.9 | 68.7 | 81.5 | 59.8 | 52.4 | 46.5 | **42.3** | 66.4 |
| *Zhyper (Ours)* | 4.2M | **74.7** | 87.2 | 85.4 | 66.0 | 68.6 | 81.0 | 59.3 | **52.6** | 44.2 | 39.6 | 65.9 |

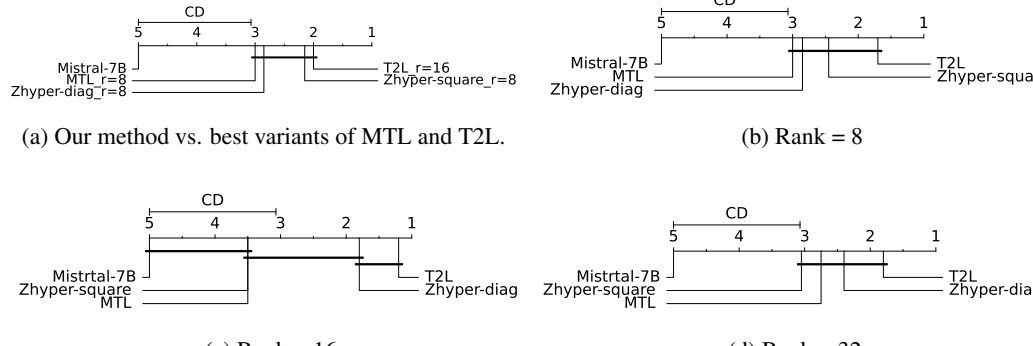

(a) Our method vs. best variants of MTL and T2L.    (b) Rank = 8

(c) Rank = 16    (d) Rank = 32

Figure 2: Critical Difference (CD) diagrams comparing our method with T2L across LoRA ranks. Lower rank is better. Unconditioned is the base model without any fine-tuning. Groups that are not significantly different are connected by a black bar.

PIQA (Bisk et al., 2019), and Boolq (BQ) (Clark et al., 2019). These benchmarks are excluded from training unless explicitly used as an oracle, and are therefore treated as unseen. Each benchmark is evaluated using three different text descriptions, and the results are averaged across them.

We compare our method against the best-performing T2L model, T2L (SFT) L with $r = 16$, which has 110 million trainable parameters. While our method does not fully match T2L's performance, it achieves comparable results while using 26x fewer trainable parameters and losing only 0.5% in the average benchmark performance (cf. Table 1); a full comparison across LoRA ranks is provided in Appendix C. To assess the significance of this difference, we apply the Friedman test followed by the post hoc Nemenyi test and visualize the results using Critical Difference (CD) diagrams. Black bars connecting different models indicate that there are no statistically significant differences w.r.t. the rank. Our analysis shows that there is no significant difference between our method, T2L, and MTL. Moreover, across LoRA ranks ($r$) 8, 16, and 32, at least one variant of our method is statistically indifferent from T2L as shown in Figure 2. Figure 3 shows that our method is on par with T2L in terms of average benchmark performance, while achieving a high parameter efficiency. The exact number

Table 2: Number of parameters.

| LoRA Rank | MTL | Zhyper-diag | Zhyper-square | T2L |
|---|---|---|---|---|
| 8 | 3.41M | 4.21M | 4.27M | 55.00M |
| 16 | 6.82M | 7.62M | 7.87M | 110.06M |
| 32 | 13.63M | 14.46M | 15.47M | 219.32M |
| **Avg. Performance** | 64.0 | 64.8 | 64.3 | 65.6 |

Table 3: Benchmark performance on unseen tasks and descriptions across layer subsets. Results are reported for Zhyper-diag with $r = 8$. Best numbers per column are in **bold**.

| Layers | Trainable Params | ArcC (acc) | ArcE (acc) | BQ (acc) | HS (acc) | OQA (acc) | PIQA (acc) | WG (acc) | MBPP (pass@1) | GSM8K (acc) | HE (pass@1) | Avg. (10 tasks) |
|---|---|---|---|---|---|---|---|---|---|---|---|---|
| all (0-32) | 4.20M | **74.7** | **87.2** | **85.4** | **66.0** | **68.6** | **81.0** | **59.3** | **52.6** | 44.2 | **39.6** | **65.9** |
| every 4th | 1.65M | 74.0 | **87.2** | 85.3 | 63.5 | 66.3 | 80.4 | 58.7 | 48.8 | 44.4 | 38.6 | 64.7 |

Table 4: Benchmark performance on unseen tasks and descriptions across embedding models. `Mistral` evaluated with Zhyper-square and `gte` evaluated with Zhyper-diag both with $r = 8$. Best numbers per column are in **bold**.

| | Trainable Params | ArcC (acc) | ArcE (acc) | BQ (acc) | HS (acc) | OQA (acc) | PIQA (acc) | WG (acc) | MBPP (pass@1) | GSM8K (acc) | HE (pass@1) | Avg. (10 tasks) |
|---|---|---|---|---|---|---|---|---|---|---|---|---|
| `Mistral` | 4.5M | 74.5 | **87.5** | 83.3 | 63.4 | **69.3** | 81.0 | **59.7** | 51.4 | 44.2 | **44.7** | 65.9 |
| `gte` | 4.2M | **74.7** | 87.2 | **85.4** | **66.0** | 68.6 | 81.0 | 59.3 | **52.6** | 44.2 | 39.6 | 65.9 |

of parameters for each method is listed in Table 2. We note that Hyperdecoders perform strongly; however, they generate a separate LoRA adapter for each problem instance, which is computationally expensive and contrasts with our approach, which generates an adapter from a text description rather than from individual problem instances. Overall, from the results of Tables 1-2 and Figure 3, we deduce that our method Zhyper offers the best trade-off between parameter-efficiency and accuracy among all considered baselines.

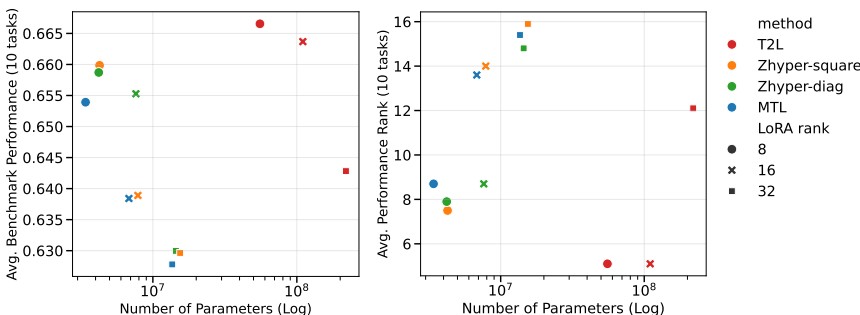

Figure 3: (Left) Average performance (higher is better); (Right) Performance rank (lower is better). Our method lies in the Pareto front optimality between performance and the number of parameters.

We conduct three ablation studies: one examining the choice of embedding model, another evaluating the number of layers adapted with LoRA, and a final one assessing the effect of the number of training datasets.

• *Embedding model ablation.* We compare `gte` with `Mistral` and observe that, across ranks and z settings, `Mistral` performs best for Zhyper-square at $r = 8$. We then compare this configuration against the best Zhyper variant using `gte` embeddings, finding that both achieve comparable performance (cf. Table 4).

• *Layer ablation.* We evaluate five configurations for Zhyper at $r = 8$, where LoRA is applied only to: the first 6 layers, the first 16 layers (i.e., the first half), the last 6 layers, the last 16 layers (i.e., the second half), and every 4th layer. Among these configurations, applying LoRA to every 4th layer yields the best performance. However, it still underperforms compared to using all layers (cf. Table 3). Selection is performed using the benchmark validation set, and final performance is reported on the test (benchmark) set. Complete tables are provided in Appendix C.

• *Training datasets ablation.* Our dataset ablation study shows that, at higher ranks and with more training datasets, the diag variant achieves better validation performance, highlighting its ability to reduce overfitting. At rank 8, the square variant performs slightly better than the diag variant, with both achieving similar performance when trained on 479 datasets (cf. Figure 4).

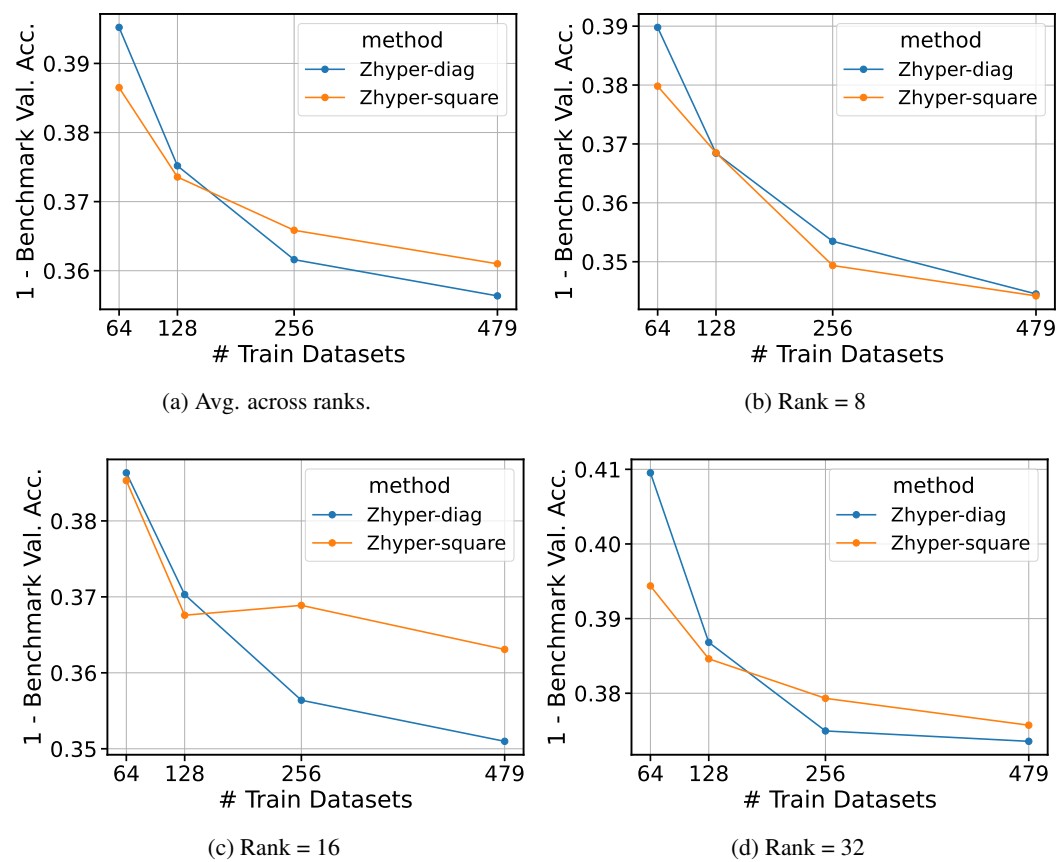

(a) Avg. across ranks.

(b) Rank = 8

(c) Rank = 16

(d) Rank = 32

Figure 4: Benchmark validation accuracy across different numbers of included training datasets.

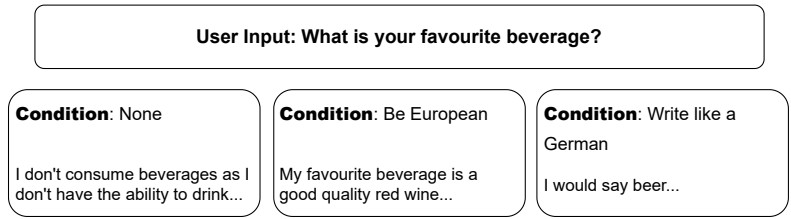

Figure 5: Model output based on text conditions. From left to right: unconditioned model, Europe-conditioned model, and Germany-conditioned model.

## 3.2 USE CASE ON CULTURAL ALIGNMENT

• *Baselines.* Similar to Section 3.1, we use `Mistral-7B-Instruct-v0.2` as our backbone and unconditional baseline. We include: *Zero-shot*, *Role-play* (prepend a short role specification to the query), *Prepending culture descriptions*, *Multi-cultural* (MTL), a single LoRA trained on either all countries or all regions, country/region-based oracle, and T2L (Charakorn et al., 2025). Additionally, we evaluate a one-hot encoding (OHE) variant of Zhyper, where the hypernetwork is conditioned on an OHE vector representing the culture.

• *Datasets.* We compile a dataset from Reddit's AskX subreddits[2]. We consider the subreddits: r/AskAGerman, r/askmexico, r/AskArgentina, r/AskTurkey, r/AskFrance, r/askegypt,

---

[2]We use Watchful1's reddit dump, which includes data between 2005-06 to 2024-12 for the top 20k subreddits (https://www.reddit.com/r/pushshift/comments/1i4mlqu/dump_files_from_200506_to_202412/)

Table 5: **Cross-country Generalization Results on CulturalBench.** We evaluate Easy/Hard settings and report accuracy (%) along with the performance rank; Cross-country generalization is assessed by partitioning countries into *seen* and *unseen* groups. Best numbers per column are in **bold**; second best are underlined; values in brackets are mean rank across countries. "N/A" indicates the setting is not applicable. All methods use a LoRA rank of $r = 8$, unless stated otherwise. Compared to prompt-based baselines and other fine-tuning baselines, Zhyper achieves the top scores on all splits and the best averages.

| | Seen Countries | | Unseen Countries | | Avg. Easy | Avg. Hard |
| | Easy | Hard | Easy | Hard | | |
| --- | --- | --- | --- | --- | --- | --- |
| Zero-shot | 58.64(6.82) | 34.95(4.96) | 53.93(5.29) | 30.48(3.76) | 55.91(5.77) | 32.36(4.13) |
| Role-play | 64.27(5.54) | 32.81(5.29) | 63.90(3.52) | 29.63(4.03) | 64.06(4.14) | 30.97(4.42) |
| Prepending culture desc. | 64.06(5.54) | 34.07(5.04) | 62.92(3.45) | 31.46(3.92) | 63.40(4.10) | 32.56(4.27) |
| Multi-cultural (MTL, $r = 16$) | 66.21(4.57) | 28.34(6.50) | 64.89(3.18) | 29.07(4.07) | 65.44(3.61) | 28.77(4.82) |
| T2L | 65.92(4.39) | 34.35(4.61) | 63.58(3.36) | 32.12(3.23) | 64.56(3.68) | 33.05(3.66) |
| Culture-specific | 67.57(4.04) | 31.84(5.25) | N/A | N/A | N/A | N/A |
| *Zhyper-OHE (Ours)* | **70.29**(2.36) | **40.58**(2.04) | N/A | N/A | N/A | N/A |
| *Zhyper (Ours)* | 70.15(2.75) | 40.39(2.32) | **67.79**(2.21) | **36.27**(2.00) | **68.78**(2.38) | **38.00**(2.10) |

r/AskAJapanese, r/AskIndia, r/AskAChinese, r/AskSouthAfrica, r/askitaly, r/AskARussian, r/AskUK, r/AskAnAmercian, r/asklatinamerica, r/AskAnAfrican, r/AskMiddleEast, r/AskEurope, r/askasia, covering 14 countries and 5 regions/continents. These subreddits were selected based on data availability. We treat each submission title and its top comment as a question-answer pair, considering the top 20k submissions and their top 3 comments based on comment score. To ensure high-quality data, we remove pairs with deleted or removed submissions or comments, as well as pairs containing references to other websites, subreddits, comments, or any type of media, following a filtering procedure similar to OpnionGPT (Haller et al., 2024). Finally, we randomly select the top 30k pairs per subreddit based on the comment score. To generate cultural descriptions, we prompt `gpt-4.1-mini` using random pairs sampled from the training dataset. Additionally, we infuse the descriptions with command-like instructions (e.g., *"Write like a German"*), so that the textual conditions reflect both stereotypical cultural traits and explicit commands to emulate the culture. We show examples and the generation prompt in Appendix E.

• *Evaluation Protocol.* We evaluate cultural alignment on CulturalBench (Chiu et al., 2025), which comprises human-written and human-verified questions spanning 45 regions and 17 topics. The benchmark provides two evaluation setups that share the same underlying questions but differ in querying format: *Easy* uses the original four-way multiple-choice questions, whereas *Hard* converts each question into four binary (True/False) statements, yielding a more challenging setting that reduces shortcutting via option heuristics. We report results at both the country and region levels[3]. Accordingly, we train two Zhyper models: one on country-level AskX data and one on region-level AskX data. For evaluation, CulturalBench questions are split into *seen* countries/regions (present in training via AskX) and *unseen* countries/regions (absent during training). For text-conditioned models (T2L and Zhyper), we use 12 cultural conditions (see Appendix E for details) to generate LoRAs per culture (country or region) and report the average performance.

• *Cultural alignment across seen/unseen countries.* As shown in Table 5, our method surpasses prompt-based approaches and fine-tuning baselines across all splits and also leads the averages. Beyond strong results on seen countries, Zhyper retains the multi-cultural compatibility that the OHE variant exhibits on seen countries, by conditioning on text, further improving transfer to unseen countries. Notably, the advantage also holds on the Hard split, indicating that the model aligns with cultural norms in a way that remains stable under stricter evaluation rather than relying on surface cues. we show an example generation in Figure 5.

• *Cultural alignment across seen/unseen regions.* Table 6 shows that Zhyper attains the best overall average at the regional level and provides balanced improvements over both seen and unseen regions, outperforming prompt-based and other fine-tuning baselines. Crucially, the margin persists on the Hard split, indicating stable regional-level gains under stricter evaluation and complementing the country-level findings under a different partition. An exception is Oceania, where competing MTL

---

[3]In this paper, *country* refers to ISO 3166-1 including administrative countries and territories, whereas *region* denotes macro-regions (e.g., North America, Middle East)

Table 6: **Cross-region generalization on CulturalBench.** We evaluate the Easy/Hard settings and report accuracy (%). Each cell is shown as Easy/Hard. Best numbers per column are in **bold**; second best are underlined. "N/A" indicates the setting is not applicable. All methods use a LoRA rank of $r = 8$, unless stated otherwise. Compared to prompt-based and other fine-tuning baselines, Zhyper shows a clear advantage on seen regions and on North America, and achieves the best overall averages.

| | Seen Regions | | | | | Unseen Regions | | Avg. |
|---|---|---|---|---|---|---|---|---|
| | Latin America | Europe | Africa | Middle East | Asia | N. America | Oceania | |
| Zero-shot | 47.52/20.79 | 56.10/31.01 | 69.40/39.55 | 45.67/21.26 | 54.41/35.08 | 67.11/40.79 | 61.54/34.62 | 55.91/32.36 |
| Role-play | 57.43/31.68 | 66.20/32.05 | 73.13/30.60 | 59.84/25.20 | 62.18/29.41 | 64.47/50.00 | 73.08/19.23 | 64.06/30.97 |
| Prepending culture desc. | 59.98/26.57 | 65.24/33.54 | 72.70/31.53 | 57.35/24.54 | 61.55/33.26 | 66.89/48.46 | 70.83/27.24 | 63.60/32.50 |
| Multi-cultural (MTL, $r = 16$) | 61.39/32.67 | 65.16/36.24 | 74.63/35.07 | 56.69/27.56 | 67.02/34.24 | 68.42/46.05 | **76.92/38.46** | 66.18/34.80 |
| T2L | 60.48/18.98 | 63.73/27.67 | 70.52/19.53 | 57.22/19.03 | 64.90/28.05 | 65.79/29.82 | 65.38/25.00 | 64.15/25.39 |
| Culture-specific | 62.38/**43.56** | 67.60/**41.11** | 77.61/39.55 | 61.42/**33.86** | 68.07/**38.03** | N/A | N/A | N/A |
| *Zhyper-OHE (Ours)* | 61.39/41.58 | **68.64**/40.77 | 75.37/**41.04** | 61.42/33.07 | **69.33**/37.18 | N/A | N/A | N/A |
| *Zhyper (Ours)* | **62.62**/35.97 | 68.23/38.78 | **78.05**/38.93 | **62.14**/29.99 | 68.79/36.40 | **71.82/53.40** | 69.23/33.65 | **68.67/37.52** |

Table 7: Delineating our method Zhyper from prior works leveraging hypernetworks (Hyperdecoder, HyperLoRA, T2L), and MTLoRA as a multi-task learning approach

| Model | Hypernetwork's Output Size | Adaptation granularity | Text-conditioned adaptation | Adapter Memory per Context |
|---|---|---|---|---|
| Hyperdecoder (Ivison & Peters, 2022) | $\mathcal{O}(Ld^2)$ | per instance | ✓ | $\mathcal{O}(Ld^2)$ |
| HyperLoRA(Lv et al., 2024) | $\mathcal{O}(rd))$ | per-context | ✗ | $\mathcal{O}(LTrd)$ |
| MTLoRA (Agiza et al., 2024) | - | shared across multiple tasks | ✗ | $\mathcal{O}(LTrd)$ |
| T2L (Charakorn et al., 2025) | $\mathcal{O}(rd)$ | per-context | ✓ | $\mathcal{O}(LTrd)$ |
| Zhyper-diag (ours) | $\mathcal{O}(r)$ | per-context | ✓ | $\mathcal{O}(LTr)$ |
| Zhyper-square (ours) | $\mathcal{O}(r^2)$ | per-context | ✓ | $\mathcal{O}(LTr^2)$ |

variants take the top performance and narrow our margin. We hypothesize that this weaker outcome reflects higher cross-regional transfer difficulty correlated with cultural divergence between Oceania and the training regions.

In both settings, Zhyper-OHE variant outperforms Zhyper on seen cultures. However, this method fails to generalize to unseen settings due to the nature of OHE.

## 4 RELATED WORK

**Low-Rank Adaptation.** To fine-tune LLMs on out-of-distribution applications, Hu et al. (2021) introduce the concept of Low-Rank Adaptation of LLMs, where the pre-trained LLM weights are frozen, and trainable rank decomposition matrices are introduced. The key concept of LoRA lies in decomposing a weight change matrix $\Delta W$ into two low-rank matrices $A$ and $B$. In Agiza et al. (2024), the authors extend the LoRA to the multi-task setting by learning shared and task-specific low-rank adapters. Ilharco et al. (2023) propose the concept of task arithmetic, where the difference between the weights of a model fine-tuned on a specific task $t$, and the base model yields a task vector $\tau_t$. This vector can then be added to another model of similar architecture to transfer task $t$. However, this approach requires a fully fine-tuned model as a reference.

**Hypernetworks.** A recent stream of research leverages Hypernetworks that build on the idea of a network's parameters being learned through another neural network (Ha et al., 2016). Hyper-Tuning (Phang et al., 2023) is used to generate LoRA weights based on a few-shot examples of a task. In Text-2-LoRA, Charakorn et al. (2025) propose a framework that performs instant adaptation of LLMs from descriptions of downstream tasks. The framework leverages hypernetworks to compress task-specific adapters and enables the zero-shot generation of new LoRA adapters at inference. Hyperdecoders are proposed in Ivison & Peters (2022) and generate task- and instance-specific decoders showing improved performance in multi-task NLP. Lastly, HyperLoRA leverages hypernetworks for generating task-specific LoRA adapters under low-rank constraints that enable efficient parameter sharing and better cross-task generalization (Lv et al., 2024).

**Discussion.** Table 7 compares Hyperdecoder, HyperLoRA, MTLoRA, T2L, and our method Zhyper along the dimension of the hypernetwork's output size, the text-conditioned adaptation and its granularity, and the adapters' memory consumption per context. We denote by $L$ the number of layers, $T$ the adapted projections (Q, V), $d$ as the hidden hidden size, and $r$ as the rank of the LoRA adapter. The key distinction is that prior methods require a hypernetwork to produce full LoRA matrices, while Zhyper introduces a compact modulation mechanism, resulting in only $\mathcal{O}(r)$ for Zhyper-diag, and $\mathcal{O}(r^2)$ for Zhyper-square, respectively. In terms of extra memory needed per context, Zhyper reduces the storage by a factor of $d$, as $r \ll d$.

While our method improves performance from simple text descriptions, LoRA-based approaches remain fundamentally limited by the capabilities of the base model. Tasks such as coding and math (e.g., MBPP, GSM8K) are inherently difficult (cf. Section 3) and typically require larger models or specialized architectures, many of which continue to struggle (Team, 2025; Team et al., 2025).

**Cultural Alignment of LLMs.** Evaluations typically use probability surveys (Haerpfer et al., 2024; Pew Research Center, 2024; Durmus et al., 2023) or non-survey suites built from authored/mined culture questions (Pistilli et al., 2024; Ju et al., 2025; Myung et al., 2024; Rao et al., 2025; Li et al., 2024b). Surveys are representative, but non-everyday questions, focusing on opinions and attitudes, are sensitive to evaluation design (Khan et al., 2025), while many non-survey suites lack rigorous validation. We adopt CulturalBench (Chiu et al., 2025) as a cultural alignment benchmark for its breadth across countries, regions, and topics and its systematic human–AI red-teaming with a challenging Easy/Hard split.

Methodologically, prior work spans anthropological/persona prompting (AlKhamissi et al., 2024), survey- or simulation-driven data curation (Li et al., 2024a;b), and distributional alignment via self-curated supervision or modified objectives (Xu et al., 2025; Yao et al., 2025; Suh et al., 2025; Cao et al., 2025). Our approach instead uses a hypernetwork to generate LoRA adapters from natural-language cultural descriptions at inference time, enabling parameter-efficient per-locale specialization with improved cross-locale generalization.

## 5 CONCLUSION

Despite the broad success of LLMs, current approaches face persistent challenges in efficiently conditioning LLMs, particularly for content alignment with a large contextual corpus. We introduce a parameter-efficient factorized hypernetwork framework, called Zhyper, for context-aware LoRA adapters given textual descriptions. Specifically, we leverage a hypernetwork that yields for each textual description a layer- and target module-specific embedding vector that is injected in LoRA adapters. Our evaluation highlights that Zhyper's computational demands are at an order of magnitude lower – up to 26x fewer parameters – compared to existing models while achieving competitive predictive performance. Through comprehensive empirical evaluation on task conditioning on 10 benchmark datasets, our method shows competitive results with state-of-the-art, while on a cultural alignment setting, Zhyper shows better generalization capabilities to out-of-domain and unseen contexts. These results highlight the potential of hypernetwork-conditioned LoRA adapters for dynamic, fine-grained LLM adaptation at minimal computational cost, supporting more sustainable and flexible model deployment.

## 6 ETHICS STATEMENT

While our method demonstrates improved cultural alignment, we acknowledge that using Reddit as a data source introduces potential biases. We do not filter the dataset for political correctness or linguistic accuracy; therefore, some QA pairs may contain harmful content. Although we select the top-voted comments, these can still be conflicting due to the diversity of users' opinions. Moreover, by relying on Reddit, we model a specific subset of people—those who use the platform—which may not accurately reflect the broader cultural perspectives of the general population.

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

## A  TRAINING PROCEDURE AND HYPERPARAMETERS

We use the following hyperparameters to train our model (Table 8). Notably, we train the model for 2,000 epochs for tasks and 5,000 epochs for cultures. For LoRA ranks below 16, training fits on a single H100 GPU (80 GB VRAM). To accelerate training, we distribute it across 8 H100 GPUs using Accelerate (Gugger et al., 2022). For example, training with LoRA rank 8 on the tasks dataset takes approximately 7–8 hours of wall-clock time, otherwise on 1 GPU, whereas on a single GPU it can take up to 48 hours.

Table 8: Hyperparameters used during training. $d_{\text{MLP\_out}}$ denotes the output dimension of the final MLP block, which serves as input to the network's output head. $d_{\text{MLP\_in}}$ denotes the input dimension of each MLP block. $d_{\text{MLP\_hidden}}$ denotes the hidden dimension of each MLP block.

| Hyperparameter | Ours/T2L | Task/Culture-specific |
|---|---|---|
| Max learning rate | 2.5e-5 | 3e-5 |
| Gradient accumulation steps | 1 | 1 |
| Batch size | 8 | 8 |
| NEFTune noise alpha | 5.0 | 5.0 |
| Warmup fraction | 0.2 | 0.1 |
| Label smoothing | 0.1 | 0.1 |
| Weight decay | 0.1 | 0.1 |
| $d_{\text{MLP\_out}}$ | 512 | N/A |
| $d_{\text{MLP\_in}}$ | 128 | N/A |
| $d_{\text{MLP\_hidden}}$ | 512 | N/A |

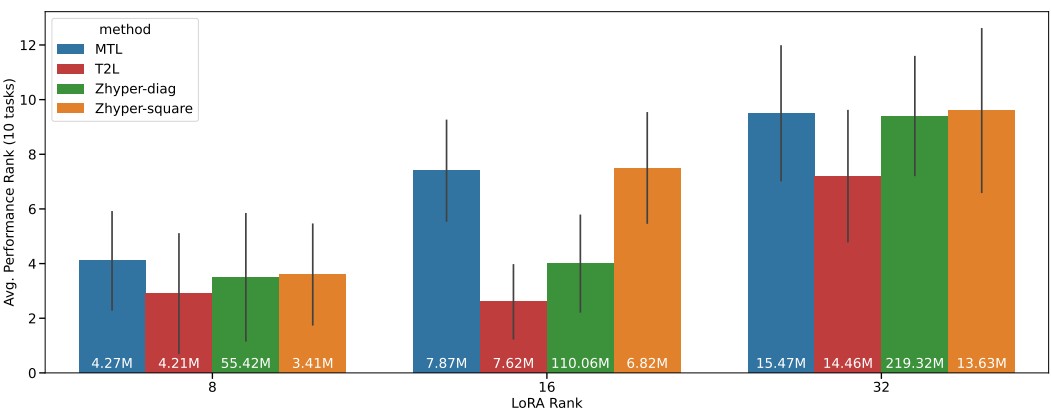

Figure 6: Average Performance Rank on the benchmark validation set (lower is better). For MTL, best variant is at $r = 8$, T2L, $r = 16$, and Zhyper $r = 8, diag$.

Table 9: SFT loss by LoRA rank $(r)$ (lower is better). Left: country models; right: region models. **Bold** indicates the best performance across LoRA ranks for each method. For Zhyper, the best variant is reported considering both the Z matrix type (diag or square) and the LoRA rank. That is, Zhyper-diag with LoRA rank 8 achieves the best performance for both country- and region-based models.

| Method | Rank | | | Method | Rank | | |
|---|---|---|---|---|---|---|---|
| | **8** | **16** | **32** | | **8** | **16** | **32** |
| MTL | 2.748 | **2.688** | 2.759 | MTL | 2.773 | **2.753** | 2.772 |
| T2L | **2.764** | 2.777 | 2.775 | T2L | **2.815** | 2.880 | 2.826 |
| Zhyper-diag | **2.705** | 2.726 | 2.734 | Zhyper-diag | **2.756** | 2.818 | 2.766 |
| Zhyper-square | **2.731** | 2.730 | 2.734 | Zhyper-square | 2.765 | 2.815 | **2.764** |

## B   HYPERPARAMETER TUNING

We report the performance of MTL, T2L, and Zhyper on a subset of the benchmark validation set (Figure 6). For Table 1, we select the best-performing variant of each method. For cultural alignment, since CulturalBench is relatively small, containing up to 200 questions per country, we do not use it as a validation set. Instead, we sample 10% of the training data (subreddit QA pairs) as a validation set and use SFT loss as the evaluation metric. The best-performing variant of each method is then used in the benchmarking tables. Table 9 reports the performance of all methods for both country- and region-based models.

## C   FURTHER ANALYSIS

We report all benchmark results for the variants evaluated on `Mistral-7B-Instruct-v0.2` in Table 12. Results for `Llama-3.1-8B-Instruct` are shown in Table 10, results for `Gemma-2-2B-Instruct` are shown in Table 11, VeRA results in Table 13, the embedding model ablation in Tables 14 and 15, and the layer ablation in Tables 16 and 17. Finally, we show a few results on LoRA-XS in Table 18.

Table 10: Benchmark performance on unseen tasks and descriptions for `Llama-3.1-8B-Instruct`. T2L and MTL results are reproduced by us, while the others are taken from T2L (Charakorn et al., 2025). Best numbers per column are in **bold**.

| | Trainable Params | ArcC (acc) | ArcE (acc) | BQ (acc) | HS (acc) | OQA (acc) | PIQA (acc) | WG (acc) | MBPP (pass@1) | GSM8K (acc) | HE (pass@1) | Avg. (10 tasks) |
|---|---|---|---|---|---|---|---|---|---|---|---|---|
| **Zero-shot adaptation without fine-tuning** | | | | | | | | | | | | |
| Llama-3.1-8B-Instruct | N/A | 73.3 | 90.6 | 80.4 | 66.6 | 75.4 | 79.8 | 55.3 | 68.7 | 75.7 | 66.5 | 73.2 |
| Prepending task desc. | N/A | 80.2 | 92.5 | 79.9 | 69.8 | 78.4 | 81.7 | 62.4 | 70.2 | 75.7 | **68.3** | 75.9 |
| **Few-shot adaptation without fine-tuning** | | | | | | | | | | | | |
| 3-shot ICL | N/A | 80.7 | 91.9 | 80.0 | 59.3 | 77.6 | 80.9 | 61.3 | 70.4 | 75.7 | 66.5 | 74.4 |
| **Zero-shot adaptation after fine-tuning** | | | | | | | | | | | | |
| MTL ($r = 8$) | 3.4M | 78.2 | 91.7 | 83.0 | 69.5 | 78.6 | 81.4 | 58.2 | 70.4 | 74.8 | 65.9 | 75.2 |
| MTL ($r = 16$) | 6.8M | 76.8 | 91.6 | 82.4 | 69.3 | 77.6 | **82.1** | 56.1 | **70.9** | 75.8 | 63.4 | 74.6 |
| MTL ($r = 32$) | 13.6M | 76.5 | 91.7 | 82.5 | 69.1 | 78.0 | 81.3 | 56.5 | 70.7 | 76.1 | 65.9 | 74.8 |
| **Conditioned zero-shot adaptation after fine-tuning** | | | | | | | | | | | | |
| T2L (SFT) L ($r = 8$) | 55.4M | **81.6** | **93.0** | **84.3** | **71.0** | 81.4 | 79.4 | 58.1 | 68.5 | 75.9 | 63.8 | 75.7 |
| T2L (SFT) L ($r = 16$) | 110.0M | **81.6** | 92.8 | 84.2 | 68.5 | 81.3 | 81.0 | 58.1 | 69.5 | 77.2 | 67.9 | **76.2** |
| T2L (SFT) L ($r = 32$) | 219.0M | 78.1 | 92.4 | 82.9 | 70.7 | 77.7 | 81.4 | 57.8 | 69.8 | 76.7 | 66.7 | 75.4 |
| *Zhyper* ($r = 8, diag$) | 4.2M | 79.5 | 92.1 | 84.0 | 70.5 | 80.0 | 78.7 | 57.4 | 69.0 | 73.2 | 62.6 | 74.7 |
| *Zhyper* ($r = 16, diag$) | 7.6M | 80.1 | 92.2 | 83.7 | 70.7 | 80.3 | 78.6 | 57.7 | 70.5 | 74.0 | 65.4 | 75.3 |
| *Zhyper* ($r = 32, diag$) | 14.5M | 79.2 | 92.5 | 82.2 | 70.2 | 78.6 | 81.4 | 57.5 | 70.0 | **77.0** | 67.1 | 75.6 |
| *Zhyper* ($r = 8, square$) | 4.3M | 79.4 | 91.8 | 83.9 | 70.2 | 79.9 | 78.8 | 57.6 | 69.8 | 74.2 | 65.4 | 75.1 |
| *Zhyper* ($r = 16, square$) | 7.9M | 80.4 | 92.3 | 83.8 | 69.7 | 79.8 | 77.7 | 57.1 | 71.3 | 75.8 | 63.2 | 75.1 |
| *Zhyper* ($r = 32, square$) | 15.5M | 77.9 | 92.1 | 82.5 | 68.7 | 77.4 | 81.6 | 57.2 | 69.8 | 76.0 | 65.2 | 74.8 |

Table 11: Benchmark performance on unseen tasks and descriptions for `Gemma-2-2B-Instruct`. T2L and MTL results are reproduced by us, while the others are taken from T2L (Charakorn et al., 2025). Best numbers per column are in **bold**.

| | Trainable Params | ArcC (acc) | ArcE (acc) | BQ (acc) | HS (acc) | OQA (acc) | PIQA (acc) | WG (acc) | MBPP (pass@1) | GSM8K (acc) | HE (pass@1) | Avg. (10 tasks) |
|---|---|---|---|---|---|---|---|---|---|---|---|---|
| **Zero-shot adaptation without fine-tuning** | | | | | | | | | | | | |
| Gemma-2-2B-Instruct | N/A | 73.3 | 89.9 | 81.0 | 55.2 | 71.0 | 71.0 | 53.8 | 12.3 | 55.6 | 43.9 | 60.7 |
| Prepending task desc. w/ ICL | N/A | 72.4 | 88.9 | 82.5 | 55.7 | 72.6 | 67.6 | 53.7 | **43.1** | 55.6 | 43.9 | **63.6** |
| **Few-shot adaptation without fine-tuning** | | | | | | | | | | | | |
| 3-shot ICL | N/A | 72.4 | 88.9 | 82.5 | 55.7 | 72.6 | 67.6 | 53.7 | **43.1** | 55.6 | 43.9 | **63.6** |
| **Zero-shot adaptation after fine-tuning** | | | | | | | | | | | | |
| MTL ($r = 8$) | 1.6M | 73.9 | 89.9 | 81.0 | 54.0 | 73.0 | **73.3** | 54.2 | 11.8 | 55.5 | 40.9 | 60.7 |
| MTL ($r = 16$) | 3.2M | 74.3 | 90.0 | 81.4 | 55.5 | 71.6 | 71.9 | 53.8 | 12.8 | 57.6 | 43.3 | 61.2 |
| MTL ($r = 32$) | 6.4M | 74.6 | 89.9 | 81.3 | **55.9** | 72.2 | 71.2 | 54.6 | 12.0 | 56.7 | **44.5** | 61.3 |
| **Conditioned zero-shot adaptation after fine-tuning** | | | | | | | | | | | | |
| T2L (SFT) L ($r = 8$) | 32.3M | 73.6 | 89.9 | 81.0 | 55.0 | 70.8 | 70.8 | 53.8 | 13.5 | 55.3 | 43.9 | 60.8 |
| T2L (SFT) L ($r = 16$) | 63.8M | 75.2 | 89.7 | 81.6 | 56.1 | 71.5 | 71.3 | 55.5 | 11.8 | 56.6 | 41.1 | 61.1 |
| T2L (SFT) L ($r = 32$) | 127.0M | **75.5** | **90.1** | 81.6 | 55.7 | **72.7** | 71.9 | 55.4 | 12.1 | 56.9 | 42.1 | 61.4 |
| *Zhyper* ($r = 8, diag$) | 2.4M | 73.8 | 89.4 | 81.4 | 52.0 | 72.3 | 72.1 | 54.0 | 12.0 | 55.3 | 37.6 | 60.0 |
| *Zhyper* ($r = 16, diag$) | 4.0M | 75.0 | 89.7 | **81.9** | 54.8 | 71.7 | 72.1 | 54.8 | 12.8 | 55.1 | 43.9 | 61.2 |
| *Zhyper* ($r = 32, diag$) | 7.2M | 74.8 | **90.1** | **81.9** | 55.4 | 72.6 | 70.0 | **55.7** | 12.4 | **57.7** | 42.7 | 61.3 |
| *Zhyper* ($r = 8, square$) | 2.5M | **75.5** | 89.6 | 81.4 | 54.6 | 72.4 | 71.2 | 54.0 | 12.8 | 55.8 | 38.2 | 60.5 |
| *Zhyper* ($r = 16, square$) | 4.3M | 73.6 | 89.9 | 81.0 | 55.0 | 70.8 | 70.8 | 53.8 | 13.5 | 55.3 | 43.9 | 60.8 |
| *Zhyper* ($r = 32, square$) | 8.2M | 73.6 | 89.9 | 81.0 | 55.0 | 70.8 | 70.8 | 53.8 | 13.5 | 55.3 | 43.9 | 60.8 |

Table 13: Benchmark performance on unseen tasks and task descriptions for `Mistral-7B-Instruct-v0.2` using VeRA (Kopiczko et al., 2024). Best overall results per column are in **bold**, while the best results for VeRA are underlined. Zhyper implementation of VeRA significantly outperforms T2L's implementation while using 3.4x less parameters (Figure 7).

| | Trainable Params | ArcC (acc) | ArcE (acc) | BQ (acc) | HS (acc) | OQA (acc) | PIQA (acc) | WG (acc) | MBPP (pass@1) | GSM8K (acc) | HE (pass@1) | Avg. (10 tasks) |
|---|---|---|---|---|---|---|---|---|---|---|---|---|
| **Standard LoRA (Hu et al., 2021)** | | | | | | | | | | | | |
| Zhyper ($r = 8, diag$) | 4.2M | **74.7** | **87.2** | **85.4** | **66.0** | **68.6** | **81.0** | 59.3 | **52.6** | **44.2** | 39.6 | **65.9** |
| **VeRA (Kopiczko et al., 2024)** | | | | | | | | | | | | |
| VeRA T2L (SFT) L ($r = 8$) | 3.43M | 69.6 | 83.7 | 73.1 | 56.2 | 57.5 | 76.6 | 52.8 | 45.3 | 41.0 | 38.0 | 59.4 |
| VeRA T2L (SFT) L ($r = 16$) | 3.43M | 67.5 | 80.5 | 71.8 | 52.6 | 54.8 | 74.4 | 49.1 | 40.7 | 40.7 | 38.8 | 57.1 |
| VeRA T2L (SFT) L ($r = 32$) | 3.45M | 67.0 | 79.2 | 71.8 | 51.6 | 55.2 | 74.7 | 47.8 | 43.9 | 41.5 | 39.2 | 57.2 |
| VeRA Zhyper ($r = 8, diag$) | 0.96M | 70.6 | 84.2 | 75.8 | 58.3 | 58.2 | 77.6 | 55.2 | 49.6 | 41.5 | 37.0 | 60.8 |
| VeRA Zhyper ($r = 16, diag$) | 0.97M | 69.1 | 83.1 | 73.6 | 56.4 | 57.1 | 77.2 | 52.4 | 46.5 | 41.0 | 37.6 | 59.4 |
| VeRA Zhyper ($r = 32, diag$) | 0.99M | 68.2 | 81.5 | 72.7 | 54.2 | 56.5 | 76.0 | 49.9 | 44.6 | 41.7 | 38.6 | 58.4 |
| VeRA Zhyper ($r = 8, square$) | 1.02M | 71.2 | 84.5 | 76.4 | 59.3 | 59.7 | 78.0 | 55.7 | 49.9 | 41.1 | 37.6 | 61.3 |
| VeRA Zhyper ($r = 16, square$) | 1.22M | 69.3 | 83.3 | 73.8 | 56.8 | 57.4 | 77.2 | 52.8 | 45.9 | 41.1 | 38.0 | 59.6 |
| VeRA Zhyper ($r = 32, square$) | 2.01M | 68.5 | 81.6 | 72.7 | 54.4 | 56.1 | 76.2 | 50.0 | 44.9 | 41.3 | 38.4 | 58.4 |

Table 12: Benchmark performance on unseen tasks and descriptions for `Mistral-7B-Instruct-v0.2`. T2L, MTL and Task-specific LoRAs results are reproduced by us, while the others are taken from T2L (Charakorn et al., 2025). Best numbers per column are in **bold**.

| | Trainable Params | ArcC (acc) | ArcE (acc) | BQ (acc) | HS (acc) | OQA (acc) | PIQA (acc) | WG (acc) | MBPP (pass@1) | GSM8K (acc) | HE (pass@1) | Avg. (10 tasks) |
|---|---|---|---|---|---|---|---|---|---|---|---|---|
| **Zero-shot adaptation without fine-tuning** | | | | | | | | | | | | |
| Mistral-7B-Instruct | N/A | 65.4 | 77.8 | 71.6 | 49.7 | 54.2 | 72.8 | 45.0 | 43.1 | 40.9 | 37.2 | 55.8 |
| Prepending task desc. | N/A | 72.0 | 85.8 | 67.6 | 58.9 | 63.4 | 77.9 | 59.0 | 41.6 | 40.9 | 39.0 | 60.6 |
| **Few-shot adaptation without fine-tuning** | | | | | | | | | | | | |
| 3-shot ICL | N/A | 72.1 | 85.9 | 71.7 | 59.0 | 66.2 | 76.2 | 58.0 | 42.6 | 40.9 | 37.2 | 61.0 |
| **Zero-shot adaptation after fine-tuning** | | | | | | | | | | | | |
| Arrow Routing ($r = 4$) | N/A | 60.9 | 86.2 | 87.6 | 80.8 | 48.6 | 83.0 | **68.5** | 50.2 | N/A | 28.7 | N/A |
| Hyperdecoders | 55.0M | 76.6 | 88.5 | 83.9 | 65.2 | 76.6 | 81.3 | 64.9 | 51.6 | 43.6 | 40.9 | 67.3 |
| MTL ($r = 8$) | 3.4M | 74.0 | 87.3 | 84.0 | 63.4 | 69.2 | 81.5 | 60.5 | 49.1 | 47.5 | 39.6 | 65.4 |
| MTL ($r = 16$) | 6.8M | 73.4 | 86.7 | 80.3 | 62.9 | 66.2 | 79.9 | 58.2 | 47.1 | 44.7 | 39.0 | 63.8 |
| MTL ($r = 32$) | 13.6M | 72.0 | 86.2 | 77.6 | 62.1 | 62.6 | 79.4 | 57.0 | 48.1 | 42.5 | 40.2 | 62.8 |
| **Fine-tuned directly on test tasks (Oracle)** | | | | | | | | | | | | |
| Task-specific LoRAs ($r = 8$) | 3.4M | 74.6 | 88.3 | **88.0** | **87.9** | **77.4** | **86.1** | 57.0 | 47.9 | **50.2** | N/A | N/A |
| Task-specific LoRAs ($r = 16$) | 6.8M | 73.6 | 87.9 | 86.9 | 84.2 | 73.4 | 84.7 | 57.1 | 47.4 | 48.1 | N/A | N/A |
| Task-specific LoRAs ($r = 32$) | 13.6M | 73.0 | 87.3 | 80.6 | 78.9 | 70.6 | 83.4 | 57.2 | 46.4 | 47.2 | N/A | N/A |
| **Conditioned zero-shot adaptation after fine-tuning** | | | | | | | | | | | | |
| T2L (SFT) L ($r = 8$) | 55.0M | 75.6 | 88.4 | 84.7 | 63.1 | 71.6 | 83.1 | 59.4 | 49.8 | 47.6 | 43.3 | **66.7** |
| T2L (SFT) L ($r = 16$) | 110.0M | 74.5 | **87.7** | 85.5 | 64.9 | 68.7 | 81.5 | 59.8 | 52.4 | 46.5 | 42.3 | 66.4 |
| T2L (SFT) L ($r = 32$) | 219.5M | 73.0 | 86.8 | 81.7 | 63.8 | 66.1 | 78.9 | 59.6 | 48.0 | 45.4 | 39.4 | 64.3 |
| Zhyper ($r = 8, diag$) | 4.2M | **74.7** | 87.2 | 85.4 | 66.0 | 68.6 | 81.0 | 59.3 | **52.6** | 44.2 | 39.6 | 65.9 |
| Zhyper ($r = 16, diag$) | 7.6M | 74.6 | 86.9 | 83.3 | 63.8 | 67.4 | 80.7 | 59.4 | 50.3 | 46.1 | **42.7** | 65.5 |
| Zhyper ($r = 32, diag$) | 14.5M | 72.0 | 86.3 | 78.1 | 62.7 | 62.4 | 79.5 | 57.5 | 47.0 | 44.2 | 40.0 | 63.0 |
| Zhyper ($r = 8, square$) | 4.3M | 74.5 | 87.4 | 83.8 | 65.1 | 69.2 | 81.6 | 58.8 | 53.8 | 45.6 | 40.0 | 66.0 |
| Zhyper ($r = 16, square$) | 7.9M | 73.2 | 86.7 | 80.4 | 61.9 | 66.3 | 79.3 | 58.9 | 49.4 | 43.8 | 39.2 | 63.9 |
| Zhyper ($r = 32, square$) | 15.5M | 71.9 | 85.9 | 77.5 | 61.7 | 62.2 | 79.2 | 58.0 | 49.2 | 43.8 | 40.2 | 63.0 |

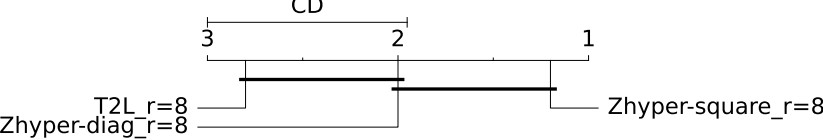

Figure 7: CD diagram comparing our method with T2L when using VeRA.

Table 14: Benchmark performance on unseen tasks and descriptions across embedding models. Best numbers per column are in **bold**.

| | Layers | Trainable Params | ArcC (acc) | ArcE (acc) | BQ (acc) | HS (acc) | OQA (acc) | PIQA (acc) | WG (acc) | MBPP (pass@1) | GSM8K (acc) | HE (pass@1) | Avg. (10 tasks) |
|---|---|---|---|---|---|---|---|---|---|---|---|---|---|
| *Zhyper* ($r = 8, diag$) | gte | 4.20M | **74.7** | 87.2 | 85.4 | **66.0** | 68.6 | 81.0 | 59.3 | **52.6** | 44.2 | **39.6** | 65.9 |
| *Zhyper* ($r = 8, diag$) | Mistral | 4.40M | 74.4 | 87.7 | 85.2 | 65.1 | **71.0** | **81.1** | 59.5 | 52.3 | 44.4 | 38.8 | **66.0** |
| *Zhyper* ($r = 16, diag$) | Mistral | 7.28M | 74.2 | 87.2 | 82.7 | 61.9 | 66.5 | 79.8 | **60.2** | 49.8 | 43.9 | 39.4 | 64.6 |
| *Zhyper* ($r = 32, diag$) | Mistral | 14.65M | 72.4 | 86.3 | 78.4 | 63.0 | 63.3 | 80.0 | 58.9 | 48.6 | 45.0 | 38.8 | 63.5 |
| *Zhyper* ($r = 8, square$) | Mistral | 4.46M | 74.5 | 87.5 | 83.3 | 63.4 | 69.3 | 81.0 | 59.7 | 51.4 | 44.2 | 44.7 | 65.9 |
| *Zhyper* ($r = 16, square$) | Mistral | 8.07M | 73.0 | 86.6 | 80.6 | 61.6 | 63.2 | 79.4 | 60.0 | 48.1 | 42.6 | 39.4 | 63.5 |
| *Zhyper* ($r = 32, square$) | Mistral | 15.67M | 71.5 | 86.3 | 77.2 | 61.5 | 61.0 | 79.1 | 58.7 | 46.7 | 43.4 | 38.8 | 62.4 |

Table 15: Benchmark validation performance on unseen tasks and descriptions across embedding models for `Mistral-7B-Instruct-v0.2`. Best numbers per column are in **bold**.

| | Layers | Trainable Params | ArcC (acc) | ArcE (acc) | BQ (acc) | HS (acc) | OQA (acc) | PIQA (acc) | WG (acc) | MBPP (pass@1) | GSM8K (acc) | HE (pass@1) | Avg. (10 tasks) |
|---|---|---|---|---|---|---|---|---|---|---|---|---|---|
| *Zhyper* ($r = 8, diag$) | gte | 4.20M | **76.8** | 86.3 | **84.6** | **61.6** | 69.2 | 80.5 | 59.5 | **52.7** | 44.5 | **39.8** | **65.5** |
| *Zhyper* ($r = 8, diag$) | Mistral | 4.40M | 75.7 | **86.4** | 84.3 | 60.5 | **69.2** | **80.6** | 59.4 | 52.0 | **44.9** | 39.6 | 65.3 |
| *Zhyper* ($r = 16, diag$) | Mistral | 7.28M | 76.4 | 86.3 | 81.9 | 58.5 | 66.9 | 79.4 | **60.1** | 49.8 | 43.8 | 39.6 | 64.3 |
| *Zhyper* ($r = 32, diag$) | Mistral | 14.65M | 74.5 | 85.5 | 77.9 | 57.6 | 63.7 | 79.6 | 58.9 | 48.9 | 44.4 | 38.8 | 63.0 |
| *Zhyper* ($r = 8, square$) | Mistral | 4.46M | 75.5 | 86.0 | 82.8 | 59.8 | 69.3 | 80.5 | 59.8 | 51.5 | 44.7 | 44.7 | 65.5 |
| *Zhyper* ($r = 16, square$) | Mistral | 8.07M | 75.7 | 86.3 | 80.4 | 57.9 | 63.6 | 78.8 | 59.6 | 48.2 | 42.8 | 39.6 | 63.3 |
| *Zhyper* ($r = 32, square$) | Mistral | 15.67M | 75.3 | 85.3 | 77.1 | 54.6 | 61.8 | 78.6 | 59.1 | 47.2 | 43.7 | 38.8 | 62.1 |

Table 16: Benchmark performance on unseen tasks and descriptions across layer subsets for `Mistral-7B-Instruct-v0.2`. Layer ranges are specified in the format [start:end:step] or [start:end]. All experiments are evaluated on $r = 8$. Best numbers per column are in **bold**.

| | Layers | Trainable Params | ArcC (acc) | ArcE (acc) | BQ (acc) | HS (acc) | OQA (acc) | PIQA (acc) | WG (acc) | MBPP (pass@1) | GSM8K (acc) | HE (pass@1) | Avg. (10 tasks) |
|---|---|---|---|---|---|---|---|---|---|---|---|---|---|
| Zhyper-square | [0:32] | 4.30M | 74.5 | **87.4** | 83.8 | 65.1 | 69.2 | **81.6** | 58.8 | **53.8** | **45.6** | 40.0 | **66.0** |
| Zhyper-diag | [0:32] | 4.20M | **74.7** | 87.2 | 85.4 | **66.0** | **68.6** | 81.0 | **59.3** | 52.6 | 44.2 | 39.6 | 65.9 |
| Zhyper-square | [0:6] | 1.50M | 71.0 | 84.0 | 75.6 | 58.6 | 62.1 | 75.0 | 57.3 | 45.5 | 42.6 | 38.6 | 61.0 |
| Zhyper-diag | [0:6] | 1.44M | 69.4 | 83.8 | 78.5 | 58.4 | 58.5 | 76.5 | 56.2 | 46.9 | 41.4 | 38.6 | 60.8 |
| Zhyper-square | [0:16] | 2.56M | 72.8 | 86.1 | 82.5 | 62.6 | 62.5 | 80.0 | 57.5 | 50.2 | 42.4 | 37.4 | 63.4 |
| Zhyper-diag | [0:16] | 2.50M | 73.4 | 86.2 | 84.6 | 63.9 | 62.0 | 80.2 | 57.5 | 51.8 | 44.3 | 39.0 | 64.3 |
| Zhyper-square | [0:32:4] | 1.71M | 73.2 | 87.2 | **85.8** | 63.0 | 63.1 | 79.7 | 58.0 | 48.8 | 45.1 | 37.8 | 64.2 |
| Zhyper-diag | [0:32:4] | 1.65M | 74.0 | 87.2 | 85.3 | 63.5 | 66.3 | 80.4 | 58.7 | 48.8 | 44.4 | 38.6 | 64.7 |
| Zhyper-square | [16:32] | 2.56M | 71.9 | 86.2 | 83.5 | 65.2 | 62.9 | 79.3 | 57.4 | 50.7 | 45.2 | 41.1 | 64.3 |
| Zhyper-diag | [16:32] | 2.50M | 72.0 | 86.3 | 83.9 | 65.8 | 62.8 | 79.6 | 57.9 | 51.4 | 46.4 | 40.4 | 64.7 |
| Zhyper-square | [26:32] | 1.50M | 72.0 | 86.3 | 81.7 | 63.2 | 60.3 | 78.7 | 56.3 | 49.5 | 47.5 | **40.9** | 63.6 |
| Zhyper-diag | [26:32] | 1.44M | 71.6 | 85.4 | 82.0 | 63.3 | 60.6 | 78.8 | 56.6 | 51.3 | 45.4 | 38.0 | 63.3 |

Table 17: Benchmark validation performance on unseen tasks and descriptions across layer subsets. Layer ranges are specified in the format [start:end:step] or [start:end]. All experiments are evaluated on $r = 8$. Best numbers per column are in **bold**.

| | Layers | Trainable Params | ArcC (acc) | ArcE (acc) | BQ (acc) | HS (acc) | OQA (acc) | PIQA (acc) | WG (acc) | MBPP (pass@1) | GSM8K (acc) | HE (pass@1) | Avg. (10 tasks) |
|---|---|---|---|---|---|---|---|---|---|---|---|---|---|
| Zhyper-square | [0:32] | 4.30M | 76.3 | **86.8** | 83.7 | 61.3 | **70.3** | 81.0 | 59.4 | **52.9** | 45.5 | 38.6 | 65.4 |
| Zhyper-diag | [0:32] | 4.20M | **76.8** | 86.3 | **84.6** | **61.6** | 69.2 | 80.5 | **59.5** | 52.7 | 44.5 | 39.8 | **65.5** |
| Zhyper-square | [0:6] | 1.50M | 72.4 | 83.5 | 76.0 | 50.1 | 59.1 | 74.6 | 57.3 | 45.6 | 42.9 | 38.6 | 60.0 |
| Zhyper-diag | [0:6] | 1.44M | 71.2 | 83.7 | 79.1 | 51.5 | 58.1 | 76.2 | 55.9 | 46.4 | 41.7 | 38.8 | 60.3 |
| Zhyper-square | [0:16] | 2.56M | 74.9 | 86.1 | 82.0 | 57.8 | 64.7 | 79.3 | 57.9 | 50.0 | 42.2 | 37.6 | 63.2 |
| Zhyper-diag | [0:16] | 2.50M | 74.5 | 85.5 | 83.2 | 58.8 | 63.9 | 79.9 | 58.1 | 51.5 | 43.9 | 38.6 | 63.8 |
| Zhyper-square | [0:32:4] | 1.71M | 75.0 | 86.4 | **84.6** | 59.1 | 64.1 | 79.3 | 58.2 | 48.9 | 44.9 | 37.8 | 63.8 |
| Zhyper-diag | [0:32:4] | 1.65M | 76.7 | 86.8 | 83.8 | 59.7 | 65.9 | 79.8 | **59.5** | 48.7 | 44.8 | 39.0 | 64.5 |
| Zhyper-square | [16:32] | 2.56M | 74.4 | 85.6 | 81.9 | 60.5 | 61.2 | 78.6 | 57.4 | 50.5 | 44.5 | 40.2 | 63.5 |
| Zhyper-diag | [16:32] | 2.50M | 74.0 | 85.3 | 82.3 | 60.9 | 60.8 | 78.7 | 58.0 | 51.2 | 46.9 | 39.8 | 63.8 |
| Zhyper-square | [26:32] | 1.50M | 73.9 | 85.9 | 80.2 | 58.9 | 58.9 | 78.2 | 56.5 | 48.8 | **47.6** | **40.4** | 62.9 |
| Zhyper-diag | [26:32] | 1.44M | 73.6 | 84.8 | 80.9 | 59.2 | 59.0 | 78.2 | 56.8 | 50.4 | 45.4 | 38.0 | 62.6 |

Table 18: Benchmark performance on unseen tasks and task descriptions for `Mistral-7B-Instruct-v0.2` using LoRA-XS (Bałazy et al., 2024). Best overall results per column are in **bold**, while the best results for LoRA-XS are underlined. LoRA-XS Performance degrades substantially at lower ranks.

| | Trainable Params | ArcC (acc) | ArcE (acc) | BQ (acc) | HS (acc) | OQA (acc) | PIQA (acc) | WG (acc) | MBPP (pass@1) | GSM8K (acc) | HE (pass@1) | Avg. (10 tasks) |
|---|---|---|---|---|---|---|---|---|---|---|---|---|
| **Base model** | | | | | | | | | | | | |
| Mistral-7B-Instruct | N/A | 65.4 | 77.8 | 71.6 | 49.7 | 54.2 | 72.8 | 45.0 | 43.1 | 40.9 | 37.2 | 55.8 |
| **Standard LoRA (Hu et al., 2021)** | | | | | | | | | | | | |
| *Zhyper* ($r = 8, diag$) | 4.20M | **74.7** | 87.2 | **85.4** | **66.0** | 68.6 | 81.0 | **59.3** | 52.6 | 44.2 | 39.6 | 65.9 |
| *Zhyper* ($r = 8, square$) | 4.30M | 74.5 | **87.4** | 83.8 | 65.1 | **69.2** | **81.6** | 58.8 | **53.8** | **45.6** | **40.0** | **66.0** |
| **LoRA-XS (Bałazy et al., 2024)** | | | | | | | | | | | | |
| LoRA-XS T2L (SFT) L ($r = 8$) | 0.86M | 0.7 | 3.5 | 1.2 | 1.0 | 0.7 | 0.4 | 0.0 | 0.0 | 0.0 | 0.0 | 0.7 |
| LoRA-XS T2L (SFT) L ($r = 16$) | 1.05M | 0.6 | 0.7 | 10.5 | 0.0 | 0.3 | 0.4 | 0.0 | 0.0 | 0.1 | 0.0 | 1.2 |
| LoRA-XS T2L (SFT) L ($r = 32$) | 1.84M | 55.7 | 71.8 | 69.1 | 3.7 | 40.3 | 62.7 | 29.1 | 2.1 | 23.0 | 19.9 | 37.7 |
| LoRA-XS T2L (SFT) L ($r = 64$) | 5.00M | 70.8 | 81.3 | 74.3 | 29.1 | 52.2 | 74.0 | 49.7 | 28.0 | 39.1 | 33.5 | 53.2 |

# D  COMPLEXITY ANALYSIS

We provide a complexity analysis of our approach compared to our competitors T2L (Charakorn et al., 2025) and HyperLoRA (Lv et al., 2024), leveraging hypernetworks with respect to the per-context materialization, their representativeness, and generalization capabilities.

**Per-Context Materialization.** For a transformer with $L$ layers and attention projections $t \in \mathcal{T}$ (e.g., $Q$, $V$), each linear map is adapted by a rank-$r$ LoRA adapter. Let $P_{\ell,t} \coloneqq r(d_{\text{in}} + d_{\text{out}})$ be the number of LoRA parameters per $(\ell, t)$-pair. The hypernetwork parameters are denoted by $P_H$.

The hypernetwork's output size is given as $\sum_{l,t} P_{l,t}$ for HyperLoRA (Lv et al., 2024) and T2L (Charakorn et al., 2025). Regarding Zhyper, it is $\sum_{\ell,t} r$ or $\sum_{l,t} r^2$ depending on the configuration -diag or -mix, respectively. In practical scenarios, we have that $r \ll d_{\text{in}}, d_{\text{out}}$, hence, $r^2 \ll r(d_{in} + d_{out})$. Therefore, in **inference**, both variants of Zhyper are far lighter than HyperLoRA and T2L. The per-context GPU memory scales as:

$$\{\text{HyperLoRA , T2L}\} \gg \text{Zhyper-square} \geq \text{Zhyper-diag} \tag{5}$$

In terms of **trainable parameters**, HyperLoRA trains $P_H + P_{\text{emb}}$ parameters, where $P_{\text{emb}}$ refers to their task query embeddings. Similarly, T2L trains on $P_H + P_{\text{layer}}(L, d_e) + P_{\text{type}}(\mathcal{T}, D_e) + P_{\text{emb}}$ parameters, i.e., layer- and type-wise embeddings are added. The learnable parameters of Zhyper aggregates to $\sum_{\ell,t} P_{\ell,t} + P_H + P_{layer}(L, d_e) + P_{type}(\mathcal{T}, D_e)$. For the models HyperLoRA and T2L, $P_H$ has to be sufficiently large such that $(A, B)$ matrices of the LoRA adapters can be generated with high fidelity. In Zhyper, we follow the idea of paying $\sum_{\ell,t} P_{\ell,t}$ once, and the hypernetwork outputs only rank-$r$ matrices as modulation signals. Therefore, in our method, $P_H$ is much smaller compared to T2L and HyperLoRA, where the hypernetwork emits $(A, B)$ directly.

**Representativeness.** Let $\mathcal{H}_{\text{full}} = \{AB : A \in \mathbb{R}^{d_{\text{in}} \times r}, B \in \mathbb{R}^{r \times d_{\text{out}}}\}$ be the hypothesis class of a LoRA adapters. That is, HyperLoRA and T2L can realize any element of $\mathcal{H}_{\text{full}}$ subject to their hypernetwork's capacity. For Zhyper-diag, we define $\mathcal{H}_{\text{diag}} = \{A \operatorname{diag}(z)B : A \in \mathbb{R}^{d_{in} \times r}, B \in \mathbb{R}^{r \times d_{out}}, z \in \mathbb{R}^r\}$ that defines a strict subset of $\mathcal{H}_{\text{full}}$. Likewise, we define $\mathcal{H}_{\text{square}} = \{AZB : A \in \mathbb{R}^{d_{in} \times r}, B \in \mathbb{R}^{r \times d_{out}}, Z \in \mathbb{R}^{r \times r}\}$ for which $\mathcal{H}_{\text{square}}$ matches $\mathcal{H}_{\text{full}}$ iff $A$ and $B$ have full row/column rank $r$. Therefore, Zhyper-square can approximate any adapter in $\mathcal{H}_{\text{full}}$. This leads to the relationship:

$$\mathcal{H}_{\text{diag}} \subseteq \mathcal{H}_{\text{square}} \subseteq \mathcal{H}_{\text{full}} \tag{6}$$

**Generalization.** Given the hypothesis classes and the number of free parameters for each of model, we have that the Rademacher complexity scales with $\mathfrak{R}(\mathcal{H}_{\text{full}}) = \mathcal{O}\left(\sqrt{\frac{r(d_{\text{in}} + d_{\text{out}})}{N}}\right)$, where $N$ is the sample size (Shalev-Shwartz & Ben-David, 2014). Likewise, we get that $\mathfrak{R}(\mathcal{H}_{\text{diag}}) = \mathcal{O}\left(\sqrt{\frac{r}{N}}\right)$ and $\mathfrak{R}(\mathcal{H}_{\text{square}}) = \mathcal{O}\left(\sqrt{\frac{r^2}{N}}\right) = \mathcal{O}\left(\frac{r}{\sqrt{N}}\right)$. This leads to the relationship:

$$\mathfrak{R}(\mathcal{H}_{\text{diag}}) \leq \mathfrak{R}(\mathcal{H}_{\text{square}}) \leq \mathfrak{R}(\mathcal{H}_{\text{full}}) \tag{7}$$

By constraining the hypothesis classes that lower the Rademacher complexity, we get tighter generalization bounds for Zhyper(-diag, or -square) compared to HyperLoRA and T2L. Notably, in practical settings with $r \ll (d_{in} + d_{out})$, the inequalities in Equation 7 become strict. Consequently, our model's performance is likely to transfer to unseen data whilst reducing the risk of overfitting and using an order of magnitude fewer parameters compared to other competitors. An empirical analysis of how training dataset size affects performance is provided in Figure 4 in Appendix C.

# E  CULTURAL CONDITIONS GENERATION

We use the following prompt with `gpt-4.1-mini` to generate culture descriptions. As a context, we append 20 QA pairs from the subreddit data. We repeat this prompt till we reach 200 descriptions.

---

**Culture Description Prompt**

You are given question–answer pairs collected from the subreddit *SUBREDDIT_NAME*. Use these pairs as background context to understand cultural attitudes.

Write 10 short and diverse descriptions of what a *NATIONALITY* person is.

You already generated the following descriptions. Please don't repeat them or generate similar ones.

*PREV_GENERATIONS*

Each description should: - Be written in plain text (no quotes or markdown).
- Use a JSON format.
- Vary in style (some short and punchy, some longer and narrative).
- Use simple, clear words so that anyone can understand.
- Do not start with "they" since it might be vague without mentioning the nationality.
- Be creative and avoid repeating the same phrasing.

Context:
*QA_PAIRS*

---

In the following, we provide examples of textual descriptions of cultural conditioning from country-(cf. E.1) and region-based (cf. E.2) perspectives. Text conditions here consists of three types: generated descriptions (e.g., *"People from Argentina tend to be curious and open to new ideas but remain cautious, preferring to understand fully before committing."*), command-like instructions (e.g., *"Think like someone from Argentina."*), and mixed forms combining both (e.g., *"Adopt Argentinian family values. An Argentinian often blends humor with seriousness, using jokes to ease tension but also to express real feelings."*). We use 128 text conditions per culture (region/country) as input for the hypernetwork.

## E.1  COUNTRY-BASED

We provide examples of textual descriptions used in our evaluation for a country-based alignment. The examples refer to the countries Argentina, France, and Japan, respectively (alphabetically ordered). For each country, we show the first four entries. We refer to our repository for an exhaustive list of textual descriptions for various countries.

---

**Argentina**

- People from Argentina tend to be curious and open to new ideas but remain cautious preferring to understand fully before committing.

- An Argentinian often uses sharp humor to cut through awkwardness, making even tense moments easier to handle.

- Think like someone from Argentina. An Argentinian often shows resilience, managing to keep going despite economic or personal setbacks.

- Adopt Argentinian family values. An Argentinian often blends humor with seriousness, using jokes to ease tension but also to express real feelings.

- . . .

---

**France**

- Many French people value practical skills and knowledge, often learning through experience and shared advice rather than just theory.
- A French person usually prefers direct and honest communication, even if it means being a bit blunt sometimes.
- A French person often values clear, logical explanations and dislikes vague or rushed answers, especially in official or professional contexts.
- In France, people often enjoy small daily rituals, like a morning coffee or a walk, as moments of calm and reflection.
- . . .

**Japan**

- Adopt Japanese daily mindset.
- Embody Japanese character.
- Many Japanese people take pride in punctuality, seeing being on time as a way to honor others' time and effort.
- Japanese individuals often enjoy seasonal celebrations but may also quietly observe traditions without much fanfare.
- . . .

### E.2    REGION-BASED

Here, we show the textual descriptions of the regions Europe, Africa, and Latin America as examples (alphabetically ordered). For each region, we show the first four entries. We refer to our repository for an exhaustive list of textual descriptions for various regions.

**Africa**

- African identity often includes a healthy dose of skepticism towards outside influence, paired with a desire to build self-reliance.
- Think like someone from Africa. An African person often carries a deep sense of resilience, shaped by a history of overcoming adversity and embracing change.
- Express African identity. Many Africans find joy in simple daily rituals, like sharing tea or storytelling at dusk, that strengthen bonds and preserve culture.
- Behave like a African local. An African person often finds strength in shared struggles, turning hardship into collective hope and determination.
- . . .

**Europe**

- Live by European principles.
- Think like a European speaker.
- Act with European mindset. Many Europeans enjoy traditional drinks with a twist, like sweet vermouth in martinis, reflecting regional tastes and history.
- Act with European reliability.
- . . .

> **Latin America**
>
> - Think with Latin American clarity.
> - Many Latin Americans find joy in street life, where music, food, and conversation create a vibrant and welcoming atmosphere.
> - Express Latin American way of life.
> - Use Latin American expressions daily. Many Latin Americans grow up with a deep respect for nature, feeling connected to the forests, rivers, and mountains that shape their daily lives.
> - . . .

# F    DETAILS OF CULTURAL ALIGNMENT EVALUATION

## F.1    DETAILS OF CULTURALBENCH

**Scope and Coverage.**    CulturalBench is a benchmark for cross-cultural knowledge and common sense. It comprises 1,696 human-written questions, each verified by five independent annotators, spanning 45 global countries as shown in Table 19, and 17 topical categories (e.g., food preferences, etiquette, festivals). We evaluate on the latest release as documented by the authors.

**Construction and Quality Control.**    Items originate from real cultural scenarios and were iteratively refined with multi-round reviewing, conflict resolution, and consistency checks to ensure unambiguous semantics and well-formed phrasing; each item includes a gold answer and brief notes to facilitate reproducibility and error analysis.

**Evaluation Setups.**    Two complementary setups are provided: **Easy** (multiple-choice) and **Hard** (the same question decomposed into binary True/False statements). These share question stems but differ in elicitation format, allowing us to assess cultural knowledge both with and without distractor options. Unless otherwise noted, we report **accuracy**. Here is an example question in the Easy and Hard setting.

*Question:* In Korean dining etiquette, what is a common practice regarding drinks and paying for the meal?

*Easy (multiple-choice).*

  **(a)** Everyone pays only for themselves.
  **(b)** Younger diners pour drinks for elders, and elders cover the bill.
  **(c)** The older person always pays, regardless of who invited.
  **(d)** The bill is typically split evenly among all diners.

*Scoring:* correct if and only if **(b)** is selected.

*Hard (binary decomposition).*

| | | |
|---|---|---|
| **(1)** Younger diners pour drinks for elders, and elders pay. | | *True* |
| **(2)** Each diner usually pays only for themselves. | | *False* |
| **(3)** Speaking loudly on entry is considered polite. | | *False* |
| **(4)** People commonly split the bill evenly. | | *False* |

*Scoring:* the item counts as correct only if all four True/False judgements are answered correctly (exact match).

**Question Template**    We follow the official CulturalBench templates. The *Easy* template (multiple choice) requires selecting exactly one option. The *Hard* template (binary question) provides a proposed answer and asks the model to select True or False.

> **Template for CulturalBench-Easy**
>
> To answer the following multiple-choice question, choose one option only among A, B, C, D.
> Instruction: You must select one option among A, B, C, D. Do not output anything else.
> Question: <Question>
> A. <Option A>
> B. <Option B>
> C. <Option C>
> D. <Option D>
> Output format: Answer: <letter>

> **Template for CulturalBench-Hard**
>
> Question: <Question>
> Answer: <Answer>
> Is this answer true or false for this question? You must choose either True or False.
> Output format: True / False

Table 19: Continents and included countries/regions in CulturalBench.

| Continent | Included Country/Region |
|---|---|
| North America | Canada; United States |
| South America | Argentina; Brazil; Chile; Mexico; Peru |
| East Europe | Czech Republic; Poland; Romania; Ukraine; Russia |
| South Europe | Spain; Italy |
| West Europe | France; Germany; Netherlands; United Kingdom |
| Africa | Egypt; Morocco; Nigeria; South Africa; Zimbabwe |
| Middle East / West Asia | Iran; Israel; Lebanon; Saudi Arabia; Turkey |
| South Asia | Bangladesh; India; Nepal; Pakistan |
| Southeast Asia | Indonesia; Malaysia; Philippines; Singapore; Thailand; Vietnam |
| East Asia | China; Hong Kong; Japan; South Korea; Taiwan |
| Oceania | Australia; New Zealand |

## F.2 CULTURAL CONDITIONS IN EVALUATION

For evaluation, we generate new culture descriptions and we make sure that they are general using the following prompt:

**Culture Description Prompt**

Write 10 short and diverse descriptions of what a *NATIONALITY* person is. Each description should:

- Be written in plain text (no quotes or markdown).

- Use a JSON format.

- Vary in style (some short and punchy, some longer and narrative).

- Use simple, clear words so that anyone can understand.

- Do not start with "they" since it might be vague without mentioning the nationality.

- Be creative and avoid repeating the same phrasing.

- Make each description sufficiently general to reflect broad patterns shared by most people from that nationality, not niche subgroups.

Output format:

- Return ONLY a JSON array of 10 objects.

- Each object MUST be: { "description": "*TEXT*" }.

Then, we construct 12 cultural conditions for evaluation by combining: (i) the 4 command prompts used during training, (ii) 4 descriptions extracted from the newly generated descriptions, and (iii) 4 new command prompts appended with 4 new generated descriptions. As illustrative examples, we show the 12 cultural conditions of Spain and Brazil, which demonstrate broad coverage and diversity.

## Spain

- Speak like a Spanish.
- Think like a Spanish.
- Act like a Spanish.
- Embody Spanish culture.
- A person from Spain often enjoys lively festivals filled with music, dance, and colorful traditions.
- Spanish people are known for their warm hospitality and love of sharing meals with family and friends.
- In Spain, many embrace a relaxed daily rhythm, including the famous afternoon siesta to rest and recharge.
- People from Spain usually speak Spanish and often have a strong connection to their local culture and history.
- A Spain native often grows up appreciating vibrant art, delicious food like tapas, and passionate football. Be a Spanish person.
- Spanish individuals typically value close relationships and celebrate life with joyful gatherings. Respond as a Spanish.
- Culturally rich, a person from Spain might enjoy flamenco music, historic cities, and outdoor cafes. Imagine you are Spanish.
- Many from Spain have a deep appreciation for outdoor living, balancing work with social time under the sun. Pretend to be Spanish.

## Brazil

- Speak like a Brazilian.
- Think like a Brazilian.
- Act like a Brazilian.
- Embody Brazilian culture.
- A Brazil person often enjoys lively music and dance, like samba and bossa nova, embracing joy in everyday life.
- Growing up in Brazil means experiencing a country full of colorful festivals, rich traditions, and warm community bonds.
- Brazil people are known for their friendly nature, welcoming smiles, and love for sharing meals with friends and family.
- A person from Brazil typically carries a deep appreciation for nature, from the Amazon rainforest to beautiful beaches.
- Many Brazil individuals have a strong passion for football, making it more than a sport but a way of connecting with others. Be a Brazilian person.
- In Brazil, people often balance modern city life with respect for cultural roots and diverse heritage. Respond as a Brazilian.
- Brazilian people usually speak Portuguese and enjoy expressing themselves through colorful clothing and vibrant celebrations. Imagine you are Brazilian.
- A Brazil person tends to have an open mind, blending influences from many cultures, creating something unique and lively. Pretend to be Brazilian.

Table 20: CulturalBench results across different types of cultural conditions. Each cell shows mean±std over the four sub-prompts under each cultural condition. Zhyper is best in all conditions on both Easy and Hard.

| | CulturalBench-Easy | | | CulturalBench-Hard | | |
|---|---|---|---|---|---|---|
| Model | Command | Description | Hybrid | Command | Description | Hybrid |
| Prepending cultural desc. | 63.39±1.81 | 63.83±0.87 | 63.57±0.45 | 31.60±1.28 | 33.05±1.38 | 32.84±0.29 |
| Text2LoRA | 64.04±0.35 | 64.22±0.28 | 64.18±0.41 | 28.20±0.89 | 23.86±0.72 | 24.12±2.02 |
| *Zhyper (Ours)* | **69.62**±0.73 | **67.95**±0.29 | **68.77**±0.29 | **38.47**±0.57 | **38.08**±0.54 | **37.45**±0.79 |

## F.3 PERFORMANCE ACROSS CULTURAL CONDITIONS

We evaluate Prepending cultural desc., Text2LoRA, and Zhyper under three cultural conditions (Command, Description, Hybrid) on CulturalBench with both Easy and Hard settings. Table 20 reports accuracy as mean±std (in percentage) aggregated over the four sub-prompts within each Cultural condition. Our method attains the best accuracy across all three cultural conditions for both Easy and Hard, while also exhibiting tight variability.

## F.4 QUALITATIVE EXAMPLES

We present two representative cases. The India example is an easy multiple-choice question whose correct option is C, garam masala and chaat masala. Zero-shot picks A, MTL outputs only ii, and T2L replies with a spice name without mapping to an option. Our method selects C, follows the required answer format, and adds a short note on how these blends are used across regional cuisines.

---

**CulturalBench-Easy Example**

**Cultural Condition**: Speak like an Indian.

**Question**
Which of the following is a commonly used Indian spice blend?

**Statements**
(i) Five spice mix     (ii) Garam masala     (iii) Pumpkin spice     (iv) Chaat masala

**Options**
A. i, iv
B. i, ii, iii
C. ii, iv
D. ii, iii

**Gold Answer**: **C** (ii, iv)

**Zero-shot**: Answer: A. i, iv. In the Indian cultural context, both five spice mix and chaat masala are commonly used spice blends. Five spice mix, also known as P̈anch Phoranïn Bengali or P̈anch Tikka Masalaïn other parts of India, is a blend of five spices: cumin, fenugreek, mustard seeds, fennel seeds, and turmeric. It is widely used in various dishes, especially in North Indian and Bengali cuisines. Chaat masala, on the other hand, is...

**Multi-cultural (MTL)**: Answer: ii

**T2L**: Garam masala

***Zhyper (Ours)***: Answer: C. ii, iv. Garam masala and chaat masala are two commonly used Indian spice blends. Garam masala is a blend of various spices that is used in Indian cooking, particularly in North Indian and Mughlai cuisine. It is typically made up of cinnamon, cloves, cardamom, cumin, coriander, and nutmeg, among other spices. Chaat masala, on the other hand, is a blend of spices that is used to season chaat, a popular Indian snack.

---

The South Africa example is a hard true/false question where the culturally correct identification is Amasi, a traditional fermented milk. MTL and T2L are also correct here, while Zero-shot confuses

near spellings such as Imasi and Amazi or generalizes to buttermilk. Beyond judging correctly, our method provides a brief cultural rationale, pointing out that Amasi is widely consumed and distinct from buttermilk and from those near-spellings; in contrast, MTL and T2L give no additional explanation.

---

**CulturalBench-Hard Example**

**Cultural Condition**: A South African usually embraces outdoor life, often passionate about nature and wildlife.

**Question**
In South African culture, which product is famously known as a traditional kind of milk or yoghurt?

**Options**
A. Imasi
B. Buttermilk
C. Amazi
D. Amasi

**Gold Judgements**: A=False; B=False; C=False; D=True.

**Zero-shot**:
A. True. Claims Imasi is a traditional fermented maize product and likens it to milk or yoghurt...
B. False. Says the traditional product is Amasi, not buttermilk...
C. False. States Amazi means rainwater; confuses terms...
D. True. Identifies Amasi as traditional fermented milk similar to yoghurt...

**Multi-cultural (MTL)**:
A. False. B. False. C. False. D. True.

**T2L**:
A. False. B. False. C. False. D. True.

*Zhyper (Ours)*:
A. False. Notes Imasi is described as a thick porridge; not a milk or yoghurt; the traditional dairy is Amasi.
B. False. Buttermilk is not the culturally specific traditional product.
C. False. Treats Amazi as a non-dairy term; the traditional dairy is Amasi.
D. True. Amasi is a traditional fermented milk widely consumed, comparable to yoghurt.

---

## F.5 RESULTS ON GLOBALOPINIONQA

**GlobalOpinionQA** (Durmus et al., 2023) aggregates multiple-choice opinion questions drawn primarily from the *World Values Survey* (WVS) and *Pew Global Attitudes* (GAS) programs (Haerpfer et al., 2024; Pew Research Center, 2024), spanning political, social, and economic themes. The benchmark contains 2,556 questions in total (2,203 from Pew; 353 from WVS Wave 7), each paired with human response distributions per country. Following the previous work, we quantify model–human agreement using the score $1 - \text{JSD}$.

To assess cross-country generalization, we partition countries into *seen* and *unseen* according to whether they appear in Ask-X data during training, and we report performance on each split as well as the overall average.

**Results.** Despite the inherently subjective nature of survey-style questions and their known susceptibility to prompt perturbations (Khan et al., 2025), averaging 12 cultural conditions yields stable estimates and reduces variance across prompts. As shown in Table 21, our method attains competitive results on both seen and unseen splits, closely tracking strong baselines while maintaining efficiency. These findings indicate that the proposed approach generalizes across countries on GlobalOpinionQA and complements the trends observed on CulturalBench.

Table 21: **Cross-country generalization on GlobalOpinionQA** We report the metric 1-JSD(Jensen-Shannon divergence). Best numbers per column are in **bold**.

|  | **Seen Countries** | **Unseen Countries** | **Avg.** |
|---|---|---|---|
| Zero-shot | 68.98 | 66.49 | 67.06 |
| Multi-cultural (MTL) | 81.87 | 80.98 | 81.18 |
| T2L | **83.64** | **82.18** | **82.52** |
| *Zhyper (Ours)* | 82.47 | 80.74 | 81.14 |

## G   LLM USAGE

In this work, LLMs were used solely as writing assistants for grammar checking, minor rephrasing, and correcting spelling or documentation in both text and code, and were not used for research ideation.

