# OpenReview forum: "Zhyper: Factorized Hypernetworks for Conditioned LLM Fine-Tuning"
_ICLR.cc/2026/Conference — Submitted to ICLR 2026_

### Official Review · Reviewer_jNbZ · 2025-10-16

**Soundness:** 3
**Presentation:** 3
**Contribution:** 2
**Rating:** 4
**Confidence:** 3

**Summary:**

This paper presents Zhyper, a parameter-efficient, factorized hypernetwork framework designed for context-aware fine-tuning of Large Language Models (LLMs). Zhyper employs a lightweight hypernetwork that generates LoRA adapters from textual inputs—such as task instructions or cultural descriptions—enabling effective adaptation to both specific tasks and cultural contexts. Compared to existing methods like T2L and HyperLoRA, Zhyper achieves this with significantly fewer parameters. Comprehensive evaluations across multiple benchmarks show that Zhyper matches or exceeds the performance of prior approaches while reducing parameter overhead by up to 26 times. It also exhibits strong generalization to unseen domains and supports fine-grained cultural alignment.

**Strengths:**

1. Unlike earlier hypernetwork-driven LoRA approaches such as T2L and HyperLoRA, Zhyper delivers on-par or superior results while drastically cutting down the number of parameters required per context, marking a significant leap in parameter efficiency.

2. By introducing a novel factorized design that separates diagonal and square modulation pathways, Zhyper advances beyond conventional hypernetworked LoRA strategies, enabling more efficient use of low-rank intermediate representations for conditional adaptation—this leads to lower memory consumption and computational load with minimal impact on model accuracy.

3. Evaluations across diverse standard benchmarks demonstrate that Zhyper consistently performs competitively with or even surpasses state-of-the-art baseline methods, highlighting its robustness and broad applicability.

**Weaknesses:**

1. While Zhyper’s core innovation, which replaces dense hypernetwork outputs with a factorized diagonal/square modulation mechanism, does lead to notable gains in parameter efficiency over predecessors like T2L and HyperLoRA, the underlying conceptual framework remains closely aligned with existing hypernetwork-based LoRA methods. The approach appears more as a refined optimization within the current paradigm rather than a transformative rethinking of how contextual signals are integrated into LLM fine-tuning.

2. The paper overlooks several key developments in efficient LLM adaptation that are directly relevant to its methodology [1,2,3,4]. Incorporating comparisons and contextualizing Zhyper against these approaches would strengthen the paper’s contribution claims and better delineate its novelty.

[1] HyperTuning: Toward Adapting Large Language Models without Back-propagation

[2] LoSiA: Efficient High-Rank Fine-Tuning via Subnet Localization and Optimization

[3] S²FT: Efficient, Scalable and Generalizable LLM Fine-tuning by Structured Sparsity

[4] LoQT: Low-Rank Adapters for Quantized Pretraining

3. The experimental evaluation is entirely based on Mistral-7B-Instruct, with no testing on larger-scale models, multilingual benchmarks, or alternative architectures outside the standard decoder-only transformer family.

**Questions:**

1.  Can the authors provide empirical evidence of Zhyper’s parameter efficiency and accuracy in LLMs larger than Mistral-7B, and non-transformer models?

2.  Beyond asymptotics, can the authors discuss or compute tighter data-dependent or task-specific generalization bounds for Zhyper-diag vs. T2L? Does reduced Rademacher complexity translate into observable gains as $n$ grows?

---

> ### Author Response · Authors · 2025-11-24
>
> > While Zhyper’s core innovation, which replaces dense hypernetwork outputs with a factorized diagonal/square modulation mechanism, does lead to notable gains in parameter efficiency over predecessors like T2L and HyperLoRA, the underlying conceptual framework remains closely aligned with existing hypernetwork-based LoRA methods. The approach appears more as a refined optimization within the current paradigm rather than a transformative rethinking of how contextual signals are integrated into LLM fine-tuning.
>
> We are thankful for the reviewer’s critical view on our work and giving us the chance to clarify some of the novelties of our approach. We agree that our approach Zhyper builds upon the hypernetwork-LoRA paradigm, however, we introduce a fundamentally different factorization approach.
> Whereas prior methods generate dense LoRA matrices (A and B) directly (cf. Table 7 in the revised draft), Zhyper learns a shared low-rank basis and generates only a context-conditioned modulation signal provided by the hypernetwork. Specifically, this decouples the contextualization from the LoRA’s adapter capacity which provides a different view on the granularity of adaptation. This is qualitatively different from T2L/HyperLoRA which rather couple generation and capacity into a single larger output space. In other words, while T2L/HyperLoRA learn a mapping from the textual descriptors to LoRA weights, our approach aims for learning a mapping from the contextual descriptors to the modulation signal.
> In terms of memory efficiency of our approach, we would like to provide a comparison in the number of FLOPs (cf. Reviewer NBTB) being executed by our approach compared to, e.g., T2L. Generally, the number of FLOPs can be estimated with $2 \cdot (d_{in} \cdot h + h^2 + h \cdot d_{out})$. Let us assume for the sake of comparability an input dimension of 1.024 and a hidden dimension of 2.048 for a mid-size LLM. The methods Zhyper and T2L distinguish in the output signal (cf. Table 7 in the revised draft). For Zhyper-diag we are resulting in approximately ~ 1.26 x 10^7 FLOPs per hypernetwork call. For Zhyper-square, we result to ~1.28 x 10^7 FLOPs. Doing the same math for T2L, where the output is $r \cdot (d_{in} + d_{out})$, we are resulting in approximately ~2.8 x 10^8 FLOPs.
> That is Zhyper (both variants) is approximately 95% cheaper compared to T2L per hypernetwork call.
> Considering the architectural design, we see it not as a minor incremental gain, but as a contribution towards a more structured, memory-efficient and modular decomposition of the adaptation process that is not provided by prior works.
>
>
> >The paper overlooks several key developments in efficient LLM adaptation that are directly relevant to its methodology [1,2,3,4]. Incorporating comparisons and contextualizing Zhyper against these approaches would strengthen the paper’s contribution claims and better delineate its novelty.
> [1] HyperTuning: Toward Adapting Large Language Models without Back-propagation
> [2] LoSiA: Efficient High-Rank Fine-Tuning via Subnet Localization and Optimization
> [3] S²FT: Efficient, Scalable and Generalizable LLM Fine-tuning by Structured Sparsity
> [4] LoQT: Low-Rank Adapters for Quantized Pretraining
>
> We thank the reviewer for pointing this out. HyperTuning (https://arxiv.org/pdf/2211.12485) is indeed closely related to our work. However, it relies on few-shot examples to train the hypernetwork, while our method trains the hypernetwork using natural language descriptions of the task or cultural context. We cite all mentioned PEFT approaches, but we refrain from direct comparison since these methods target different settings. For example, LoQT is designed for quantized models, and S²FT and LoSiA focus on selecting structured subsets of model parameters for training. Our work instead focuses on LoRA (and recent variants) in standard fine-tuning scenarios, which makes the above methods less directly comparable to our setting.
>
> > Can the authors provide empirical evidence of Zhyper’s parameter efficiency and accuracy in LLMs larger than Mistral-7B, and non-transformer models?
>
> Thank you for the suggestion. We agree that evaluating Zhyper on non-transformer architectures is an interesting direction for future work. However, due to limited time and resources, we were only able to extend our experiments to additional transformer-based LLM families, including Llama-3.1-8B-Instruct (Table 10), and Gemma-2-2B-IT (Table 11).

---

> ### Author Response · Authors · 2025-11-24
>
> > Beyond asymptotics, can the authors discuss or compute tighter data-dependent or task-specific generalization bounds for Zhyper-diag vs. T2L? Does reduced Rademacher complexity translate into observable gains as n grows?
>
> In our empirical study, we evaluate our models only on unseen datasets. To fulfill the reviewers request about the behavior of increasing $n$, we report the performance on the validation accuracy as an empirical proxy for the generalization performance. While the Rademacher complexity does not directly bound the absolute validation accuracy, it justifies why a lower-capacity variant such as the Zhyper-diag remains competitive. As shown by Eq.7 the Rademacher Complexity ensures that models having a smaller hypothesis class have a tighter worst-case generalization guarantee than models in larger classes. With that in mind, Zhyper-diag is less prone to overfitting in the worst case. We report the validation performance on increasing n as a new study in Figure 4. Zhyper-diag’s procedure on just scaling the latent directions of the A/B matrices of the LoRA adapters provides a form of regularization that results in superior generalization capabilities compared to Zhyper-square.
>
> > We believe we have addressed all the concerns raised by the reviewer. We invite the reviewer to consider their score accordingly

---

### Official Review · Reviewer_cgzs · 2025-10-31

**Soundness:** 3
**Presentation:** 3
**Contribution:** 2
**Rating:** 6
**Confidence:** 4

**Summary:**

This paper proposed a new method for parameter efficient finetuning.
The key idea is that the Lora weight for making a model meet a new requirement can be generated from the textual description using a hypernetwork.
The method generate the parameters for the Lora adapter by using an embedding that can tell the contextual information, the layer number and the attention module for the lora.
Experiments show that the proposed method can generate useful weights for LLMs to transfer to unseen tasks or align with an unseen culture during the hypernetworks training.

**Strengths:**

1. The task is an important one, and the paper proposed method seems to be effective.
2. The proposed method can save a magnititude of number of parameters compared to similar methods.

**Weaknesses:**

1. The experiments should try to cover a wide range of base LLM to show the proposed method is not just useful for one base LLM.
2. Would reasoning LLM benefit from this method?
3. The method only modifies a very small part of the base LLM, thus I think the models abilities are still bounded by the base LLM. It would be great if the paper can discuss on that and maybe show some negative results that the proposed method and other method cannot help base LLM learn tasks beyond the base LLM.

**Questions:**

See weaknesses

---

> ### Author Response · Authors · 2025-11-24
>
> >The experiments should try to cover a wide range of base LLM to show the proposed method is not just useful for one base LLM.
>
> Thank you for the suggestion. We have extended our experiments to include Llama-3.1-8B-Instruct (Table 10) and Gemma-2-2B-IT (Table 11).
>
> >Would reasoning LLM benefit from this method?
>
> We believe that any LLM, including reasoning ones, would benefit from this method. However, due to the limited time available during the rebuttal phase, we were unable to conduct additional experiments to empirically validate this claim. We view this as a promising direction for future work and plan to explore it in subsequent revisions.
>
> > The method only modifies a very small part of the base LLM, thus I think the models abilities are still bounded by the base LLM. It would be great if the paper can discuss on that and maybe show some negative results that the proposed method and other method cannot help base LLM learn tasks beyond the base LLM.
>
> We would like to thank the reviewer for an interesting point. The limitation described in the reviewer’s comment applies to all LoRA-based and PEFT-based adapter methods and is not unique to Zhyper. Every PEFT method keeps the parameter of the base model fixed and only adds a low-rank update in the sense of $W’ = W_{\text{base}} + \delta W$. This means that we cannot provide fundamentally new knowledge that exceeds the internal representation of the base model, and therefore, LoRA adapters do not aim at providing *entirely* new capabilities going beyond what the frozen layers include. LoRA only modifies the directions already latent in the pretrained model. Therefore, we agree with the reviewer that LoRA approaches are by theirs nature in a certain degree limited to the expressive power of the base LLM.
> The approach we are focusing on - and the models of our competitors – is how these LoRA matrices are impacted in terms of low-rank corrections on top of the frozen base weights of the LLM. Specifically, Zhyper adapts the weights conditioned by any contextual information, e.g., cultural descriptions.
> In our approach, we provide two approaches, namely Zhyper-diag and Zhyper-square, where the latter tries to provide more expressiveness. Specifically, the -diag variant provides a scaling/masking on the latent directions of the A/B matrices of a LoRA adapter - comparable to VeRA - , whereas the latter relaxes the operations on the matrices by allowing for a mix of the latent directions.
> Therefore, we agree with the reviewer that the limitation of LoRA adapters is that they cannot give an LLM capabilities beyond the base model as they operate as low-rank corrections on top of frozen base weights. However, Zhyper is more parameter-effiient than a full LoRA approach. To be precise about these limitations, we added a paragraph in the ‘Discussion’ part of Section 4.
>
>
>
> > We believe we have addressed all the concerns raised by the reviewer. We would kindly ask the reviewer to increase the score based on the provided clarifications.

---

### Official Review · Reviewer_W8mz · 2025-10-31

**Soundness:** 3
**Presentation:** 3
**Contribution:** 2
**Rating:** 4
**Confidence:** 3

**Summary:**

Conditioning Large Language Models (LLMs) to align with specific textual descriptions, such as tasks or cultural values, often introduces a large number of parameters when using existing hypernetwork-based LoRA methods. This paper proposes Zhyper, a parameter-efficient factorized hypernetwork framework that generates a compact, context-aware modulation signal z from these textual descriptions. Instead of generating the full adapter weights, Zhyper injects this small z signal between shared, trainable LoRA matrices A and B to compute the final weight update. Experiments show Zhyper achieves competitive performance on task-conditioning benchmarks with up to 26x fewer parameters than baselines, while also demonstrating improved generalization in the novel use case of cultural alignment.

**Strengths:**

* This paper proposes ZHYPER, which enables efficient conditional generation while significantly reducing the number of parameters compared to previous methods.
* Experimental results show that ZHYPER achieves fine-tuning performance comparable to larger models while using only one-tenth of the parameters.
* The authors provide a theoretical analysis demonstrating the superior generalization ability of ZHYPER.
* The paper is well-written.

**Weaknesses:**

* Overall, this work presents an improvement over T2L. Instead of generating the entire LoRA, it only needs to generate a low-rank embedding. However, the contribution is still largely incremental.
* The experiments are conducted on too few models. The authors only evaluate on Mistral-v0.2, lacking experiments on a wider range of models such as Qwen3, Llama3, and Gemma3. I believe the authors should test across different model families and scales.
* There is no ablation study on the embedding model. Since all experiments use the same embedding model, the authors should perform ablation experiments to examine its influence. Additionally, what would happen if the model’s own embedding outputs were used as inputs?
* The paper lacks comparison with Task Vector, which serves a similar purpose by injecting domain-specific information.

**Questions:**

* Is it necessary to apply LoRA to every layer? Would applying it only to certain layers lead to better performance?

* Could the authors provide experiments with more embedding models and large language models (LLMs)?

* Could the authors compare their method with the Task Vector approach and clarify the differences between the two?

* Why does the method generate fewer parameters? Since both approaches use LoRA, the total number of additional model parameters is not reduced — only the number of generated parameters is smaller.

---

> ### Author Response · Authors · 2025-11-24
>
> > Is it necessary to apply LoRA to every layer? Would applying it only to certain layers lead to better performance?
>
> Thank you for pointing this out. We performed an ablation study across several configurations, including selecting every fourth layer, the last 6 layers, the last 16 layers (i.e., the last half), the first 6 layers, and the first 16 layers (i.e., the first half) using Zhyper LoRA (diag). We have added the comparison results in Table 3.
>
> > Could the authors provide experiments with more embedding models and large language models (LLMs)?
>
> Thank you for your suggestion. We included experiments using Mistral as an embedding model (Table 4), experiments for Llama-3.1-8B-Instruct in Table 10 and Gemma-2-2B-IT in Table 11.
>
>
> > Could the authors compare their method with the Task Vector approach and clarify the differences between the two?
>
> Thank you for the insightful suggestion. We reviewed the Task Vector approach (e.g., Ilharco et al., 2022, https://arxiv.org/pdf/2212.04089), which constructs task vectors by subtracting the weights of an untrained (or base) model from those of a model fully fine-tuned on a specific task. These task vectors can then be linearly combined and added to a model’s weights to achieve multi-task adaptation or even task analogies, where combining vectors from several tasks enables adaptation to a related, unseen task.
> However, the Task Vector methodology relies on having fully fine-tuned models for each task to compute these weight differences. In contrast, our work focuses exclusively on parameter-efficient fine-tuning (PEFT), specifically Low-Rank Adaptation (LoRA), where only a small set of low-rank parameters are trained rather than the full model. Because task vectors require full-model fine-tuning, they are not directly applicable within our LoRA-based setting. For this reason, the Task Vector approach is not compatible with our problem formulation. We compare Task Vectors to our setting in the Low-Rank Adaptation subsection of Section 4.
> Please let us know if there are any additional comparisons or clarifications that would be helpful.
>
> > Why does the method generate fewer parameters? Since both approaches use LoRA, the total number of additional model parameters is not reduced — only the number of generated parameters is smaller.
>
> During training, the hypernetwork learns to generate a small matrix (in the hyper-square variant) or a vector (in Zhyper-diag) denoted Z, while the LoRA matrices A and B are trained in the usual way. In contrast, T2L-L generates full LoRA matrices A and B using a much larger hypernetwork with a head of size $d_{MLP\_out}\ \times r \times (d_{out} + d_{in})$ Our method requires only a smaller hypernetwork with a head of size $d_{MLP\_out}\ \times r$ that outputs Z. Because our hypernetwork is smaller, our method uses fewer parameters during both training and inference. This reduction does not change the overall size of the original base model.
> All parameter counts in the tables include every component used during training, including the LoRA matrices A and B as well as the hypernetwork.
>
>
> > We believe we have addressed all the concerns raised by the reviewer. We invite the reviewer to consider their score accordingly

---

### Official Review · Reviewer_NBTB · 2025-11-01

**Soundness:** 2
**Presentation:** 1
**Contribution:** 1
**Rating:** 2
**Confidence:** 4

**Summary:**

The paper is built on top of Text-to-LoRA (T2L). T2L trains a hypernetwork that generates a task-specific LoRA matrix given a textual task description. This paper proposes two architecture modifications that reduce the number of the parameters of the hypernetwork used to generate task-specific LoRA. The proposed model performs competitively with SOTA methods with reduced number of parameters.

**Strengths:**

- Thorough empirical comparison with existing baselines on various tasks
- The architecture is simple and effective at reducing the number of parameters

**Weaknesses:**

- This paper misses important prior works, i.e., LoRA-XS [2], VeRA [3], which propose exactly the two parameterizations used in this work (square and diag). The existence of these two prior works directly dilutes the contributions of the paper, making it hard to justify acceptance as the paper's core idea is simply chaning the output space of T2L from LoRA to LoRA-XS or VeRA without much insights gained.
- This paper focuses specifically on reducing the number of parameters of the hypernetwork. However, the cost (time and memory) of running a forward pass over the hypernetwork is a fraction of generating a response done by the target LLM. Therefore, I respecfully doubt that the propose achitecture would impact any meaningful metrics related to LLM inference (e.g., VRAM, latency, etc.)
- Section 2.1 and 2.2 are largely based on T2L [1] but is not explicitly mentioned in the text, which makes it hard to identify the novelty of this paper. I believe that by explicitly mentioning how T2L formalizes the problem and what its architecture looks like in Section 2.2 would be beneficial for reader's understanding and scientific credit assignment.
- Various notation mistakes, causing significant confusion (see Typos and Notation Mistakes)
- Table 5 is highly subjective without any supporting empirical evidence

#### Typos and Notation Mistakes
- $r$ is reused with different meanings in line 107 and 108
- At line 108, the hypernetwork maps a layer-specific description vector to the entire parameters of the target network
- $H_\phi$ is defined three times with different meanings (line 108, 130, and 142)
- $z$ is defined with two different meanings (line 108 and 132)
- $\theta$ is defined with two different meanings (line 109 and 146)
- $j$ is not defined in Eq. 4 making it illegible
- line 175: 'HyprLoRA' should be 'HyperLoRA'
- 'Zhyper-full' is never defined but used at line 186
- line 460-461; opening parentheses should have a preceding white space

Overall, I believe that there are several major concerns in the current version of the submission and, therefore, I recommend reject.

[1] Charakorn, Rujikorn, et al. "Text-to-LoRA: Instant Transformer Adaption." ICML 2025

[2] Bałazy, Klaudia, et al. "Lora-xs: Low-rank adaptation with extremely small number of parameters." arXiv preprint 2024

[3] Kopiczko, Dawid J., Tijmen Blankevoort, and Yuki M. Asano. "Vera: Vector-based random matrix adaptation." ICML 2024

**Questions:**

- What does "conditioning the LoRA weights" (line 17) mean?
- What does "large contextual spaces" (line 50) mean?
- How come the hypernetwork can "induce descriptive information" (line 94)?

---

> ### Author Response · Authors · 2025-11-24
>
> We would like to thank the reviewer for the valuable feedback. Below we will address the concerns of the reviewer:
>
> > This paper misses important prior works, i.e., LoRA-XS [2], VeRA [3], which propose exactly the two parameterizations used in this work (square and diag). The existence of these two prior works directly dilutes the contributions of the paper, making it hard to justify acceptance as the paper's core idea is simply chaning the output space of T2L from LoRA to LoRA-XS or VeRA without much insights gained.
>
> Thank you for raising this point. VeRA (https://arxiv.org/pdf/2310.11454) trains two diagonal matrices, $\lambda_b$ and $\lambda_d$ (cf. Section 3.1).
> Likewise to Zhyper-diag, it provides a scaling of the LoRA space, where VeRA provides the additional option to scale them independently. Therefore, VeRA provides a multiplicative rescaling of each LoRA direction stored in the matrices A and B, separately. In comparison to VeRA, our framework also allows to mix the r latent directions through the full (r x r)-matrix provided by Zhyper-square. Therefore, our method is still cheap compared to full LoRA, but gains expressiveness through the re-combination of learned LoRA basis functions.
> A T2L version of VeRA would generate only $\lambda_b$ and $\lambda_d$ matrices for each task, whereas our approach introduces an additional matrix Z that is trained jointly with them. At inference time, $\lambda_b$ and $\lambda_d$ are computed once and remain fixed, and only Z is produced for each new task. In contrast, T2L must regenerate $\lambda_b$ and $\lambda_d$ for every task, which requires a much larger hypernetwork. As shown in Table 12, our method substantially reduces the number of trainable parameters while outperforming T2L VeRA.
> Our Zhyper-square variant is closely related to LoRA-XS. However, in LoRA-XS the LoRA matrices are initialized via SVD and then frozen, whereas in our setting they are trained jointly with Z. In addition, LoRA-XS provides only a square-matrix variant, while we also propose a diagonal variant. We are currently running experiments to empirically compare these settings and aim to include the results before the end of the rebuttal phase.
>
> > This paper focuses specifically on reducing the number of parameters of the hypernetwork. However, the cost (time and memory) of running a forward pass over the hypernetwork is a fraction of generating a response done by the target LLM. Therefore, I respecfully doubt that the propose achitecture would impact any meaningful metrics related to LLM inference (e.g., VRAM, latency, etc.)
>
> Thanks to the reviewer for pointing to the computational overhead of our approach. We agree with the reviewer's correct observation that the total number of FLOPs of the hypernetwork's generation are only a small amount <0.1% of a full LLM forward pass. Hence, it is beneficial to mention that even the memory footprint is of negligible amount.
>
> To be concrete, the motivation of our work is to provide a cheaper fine-tuning mechanism and to provide an approach for context-aware conditioning with a very compact modulation signal provided by the hypernetwork.
> Speaking of the compactness of our approach, we can compare FLOPs being roughly $2 \cdot (d_{in} \cdot h + h^2 + h \cdot d_{out})$. Let us assume for the sake of comparability an input dimension of 1.024 and a hidden dimension of 2.048 for a mid-size LLM. The methods Zhyper and T2L distinguish in the output signal (cf. Table 7 in the revised draft). For Zhyper-diag we are resulting in approximately ~ 1.26 x 10^7 FLOPs per hypernetwork call. For Zhyper-square, we result to ~1.28 x 10^7 FLOPs. Doing the same math for T2L, where the output is $r \cdot (d_{in} + d_{out})$, we are resulting in approximately ~2.8 x 10^8 FLOPs.
> That is Zhyper (both variants) is approximately 95% cheaper compared to T2L per hypernetwork call.
> Generally, Zhyper's signal tries to avoid generating full matrices, i.e., millions of parameters per user context. That being said, even though the saving might not be as relevant for a single-user inference, it is meaningful for large scale scenarios where we deploy LLM as a service with many concurrent users. In the scope of sustainable AI, reducing the number of parameters that provide eco-friendly infrastructure is a fundamental task.

---

> ### Author Response · Authors · 2025-11-24
>
> > Section 2.1 and 2.2 are largely based on T2L [1] but is not explicitly mentioned in the text, which makes it hard to identify the novelty of this paper. I believe that by explicitly mentioning how T2L formalizes the problem and what its architecture looks like in Section 2.2 would be beneficial for reader's understanding and scientific credit assignment.
> Thank you very much for pointing on this important and intersting related work. We agree with the reviewer that both approaches have similar settings. Concretely, both propose hypernetwork-based methods for conditional LoRA generation.
> However, they differ in the core objective. Whereas T2L’s goal ist to generate full LoRA adapters (-large variant) from natural-language task descriptions, the goal of Zhyper is rather to provide a mechanism for conditioning an LLM’s behavior using modulation signals instead of generating full LoRA matrices. Hence, Zhyper yields compact modulation vectors that scale the vectors from the A/B matrices or mix them for Zhyper-square, respectively.
> Moreover, the models differ in how to solve a task. Whereas T2L focuses on task adaptation, Zhyper’s goal is task alignment which brings both methods naturally in comparable situations where the settings align with each other. Whereas T2L’s adaptation is encouraging the model to solve a task by learning the right skill set, Zhyper’s goal is to steer a model’s behavior on a task to follow certain norms, preferences or even constraints.
>
> We follow the reviewer’s suggestion on that point and highlight the discussed differences in more detail to position our paper more concrete in the research field and strengthen the delineation from existing works.
>
>
> > Various notation mistakes, causing significant confusion (see Typos and Notation Mistakes)
>
> We appreciate pointing at the room for improvements regarding the notation. To be specific, we revised them accordingly to differentiate between the recap on the hypernetworks and the sections introducing Zhyper-diag and Zhyper-square, respectively. We kindly refer to ‘Typos and Notation Mistakes’ for further details about the updates in the notation.
>
> > Table 5 is highly subjective without any supporting empirical evidence
>
> We thank the reviewer for pointing at Table 5 (in the revised draft Table 7). To be more accurate in the delineation of our approach to existing works, we revised the table accordingly.
> In the updated version, we are precisely stating the hypernetwork’s output size and how it compares to existing work. Specifically, Hyperdecoder generated adapter weights in the size of $d^2$ for all layers L, resulting in $\mathcal{O}(Ld^2)$. HyperLoRA and T2L(-large) have output dimensions of $\mathcal{O}(rd)$ for generating LoRA weights. Notably, we reduce in our approach the output dimension further, resulting in $\mathcal{O}(r^2)$ for the -square variant, and only $\mathcal{O}(r)$ for the -diag variant.
> Moreover, we now provide the reader a comparison of the adapter’s extra memory per context, resulting for Hyperdecoder to $O(Ld^2)$. For HyperLoRA, MTLoRA, T2L we state $O(LTrd)$ for $r \times d$ being the output dimension for $L$ layers and $T$ modules (Q, K). In comparison, our approach stands out with Zhyper -diag resulting in $\mathcal{O}(LTr)$ and $\mathcal{O}(LTr^2)$ for Zhyper-square, respectively.
> In practice, we have that $d \gg r$ providing a more parameter-efficient hypernetwork training for Zhyper whilst having a reduced memory footprint and showing competitive generalization capabilities..
>
> > Typos and Notation Mistakes
>
> Thanks for pointing out the typos and notation flaws.
> Specifically, the general general introduction of the hypernetwork in line 108 now uses vector $v$ as descriptor variable instead of $z$ to differentiate between the general formalization and our usage of the descriptor variable $z$, as well as parameter $\phi$ instead of $\theta$ to avoid the clash with Eq. 4.
> Moreover, to differentiate between the hypernetworks, we introduce:
>
> - Zhyper-diag: $H^{vec}_{\phi}$
>
> - Zhyper-square: and $H^{sq}_{\phi}$.
>
> The $j$ in Eq. 4 was introduced in line 125, however, we recapped it again specifically after Eq. 4 for clarification, where $c_i^{(j)}$ denotes the “the $j$-th contextual descriptor fo the $i$-th dataset.”
>
> > We believe we have addressed all the concerns raised by the reviewer. We invite the reviewer to consider their score accordingly

---

> ### Comment · Reviewer_NBTB · 2025-11-26
>
> Thank you the authors for providing the rebuttal and updating the paper.
>
> - I still believe that the diag variant is equivalent to using the $\lambda_d$ diagonal matrix from the VeRA paper while the square variant is equivalent to the LoRA-XS paper. The fact that the A and B matrices are both trained is not a novel innovation but rather small distinction that does not warrant publication.
> - Thank you the authors for providing the additional analysis. Since the forward pass is so negligible, I would suggest the authors to frame the reduced parameters as a way to speed up training instead. I imagine that this framing would be more impactful given that the hypernetwork will be trained for multiple gpu-days.
> - I still find the revised paper to be somewhat hard to read even after having read it before.
>
> Overall, I thank the authors for improving the paper but I stand that the current submission does not meet the publication standard of ICLR and I cannot recommend acceptance.

---

> > ### Author Response · Authors · 2025-11-28
> >
> > We thank the reviewer again for the constructive feedback regarding our paper. We appreciate the reviewer’s time for the overall assessment during the rebuttal phase.
> >
> > > I still believe that the diag variant is equivalent to using the diagonal matrix from the VeRA paper while the square variant is equivalent to the LoRA-XS paper. The fact that the A and B matrices are both trained is not a novel innovation but rather small distinction that does not warrant publication.
> >
> > We would respectfully like to correct the reviewer’s assessment about the model’s behavior, as the (i) equivalence of Zhyper-diag to the VeRA approach and the (ii) equivalence between Zhyper-suqare to LoRA-XS do not factually hold. Let us discuss why:
> >
> > (i) Zhyper-diag and VeRA
> > To recap on VeRA’s workflow, the update rule is defined as: $h = W_0 x + \Delta W x = W_0 x + \Lambda_b B \Lambda_d A x$. Therefore, VeRA has two diagonal matrices, one for $A$ and one for $B$, whereas our approach just uses one. Thus, it does not hold that Zhyper-diag is equivalent to VeRA. I.e.:
> > VeRA: $\Lambda_b B \Lambda_d A$ with frozen $A, B$; learnable $\Lambda_b$ and $\Lambda_d$
> > Zhyper: $A Z B$ with trainable $A, B$; and one learnable $Z$ (for -diag: learnable vector $z$ that is diagonalized)
> >
> > (ii) Zhyper-square and LoRA-XS
> > LoRA-XS is defined as $\Delta W= U_r \Sigma_r R V_r^T$. Therefore, it applies a truncated SVD formulation for $A$ and $B$. Thus, Zhyper-square is *not* equivalent to LoRA-XS. Specifically, in LoRA-XS, we have $A =  U_r \Sigma_r$ and $B=V_r^T$, where the matrices $U$ and $V$ contain the left/right singular vectors corresponding to the top $r$ values. LoRA-XS builds on a per-task trained matrix $R$.
> >
> > In addition to the above major factorization changes, we highlight that Zhyper is a zero-shot PEFT approach that does not need training for a new task, while VeRA and LoRA-XS need to be trained on each new task.
> >
> > Not only is our approach methodologically different from simply adopting T2L with VeRA or LoRA-XS factorizations, but it also leads to significantly better empirical performance in zero-shot PEFT, outperforming T2L+VeRA (Table 13).
> >
> > We replicated the implementation of LoRA-XS with $A=U_r \Sigma_r$, B=$V_r^T$ and $\Delta W= U_r \Sigma_r R V_r^T$ and froze them while using the hypernetwork to produce the $R$ matrix. Under this setup, we observed a substantial degradation in model capability: outputs frequently became nonsensical, resulting in performance worse than the base model in most cases. We have updated our repository with the LoRA-XS code and report the corresponding results in Table 18.
> >
> > > Thank you the authors for providing the additional analysis. Since the forward pass is so negligible, I would suggest the authors to frame the reduced parameters as a way to speed up training instead. I imagine that this framing would be more impactful given that the hypernetwork will be trained for multiple gpu-days.
> >
> > While reduced parameter count can indeed translate to faster training, our goal in emphasizing parameter efficiency is its connection to generalization, rather than speed alone. We have provided a theoretical understanding of this advantage in our ‘Complexity Analysis’ (moved to the appendix; see reviewer’s third concern). Additionally, we demonstrated the empirical advantage of our method thoroughly in Section 3.
> > The ability to generalize better is a main attribute for Machine Learning methods, and prediction models that can achieve a strong performance with a reduced Rademacher Complexity are always preferred. As a result, we believe the benefit of our method is evident.
> >
> >
> > > I still find the revised paper to be somewhat hard to read even after having read it before.
> >
> > Thanks for pointing out the reading difficulties again. To further enhance the readability of our work, we moved the chapter about the complexity analysis to the appendix, which is intended for further insights for the interested reader, but should not distract from the main focus. We believe that this further helps in grasping the main idea of our work. Instead, we moved more ablation studies to the main content, which highlights our model’s performance w.r.t. the number of training datasets used as input.
> > We fixed all the notation suggestions of the reviewer. We would be happy if the reviewer could point to any direct improvement we can further consider.
> >
> >
> >
> > > Considering that we concretely demonstrated that the reviewer’s concerns arise from a misunderstanding of the differences to VeRA and LoRA-XS, and considering the provided empirical evidence, we invite the reviewer to consider their score accordingly.

---

### Meta-Review · Area_Chair_ry5o · 2026-01-13

**Summary:**

Reviewers acknowledged the parameter efficiency of the method and the extensive additional experiments provided during the rebuttal. However, the lack of fundamental novelty distinguishing it from the combination of existing methods prevents it from standing out as a significant contribution to the ICLR community at this time. The authors are encouraged to further differentiate their work theoretically or demonstrate unique capabilities enabled only by this specific factorization in future submissions.

**Reviewer Concerns:**

1: A primary and persistent criticism is that the proposed method is largely incremental. While the authors clarified differences in training setups, the reviewers maintained that these are minor distinctions.

2: Despite the authors' efforts to correct typos and reorganize sections during the rebuttal, the Reviewer NBTB explicitly stated that the revised manuscript remained "somewhat hard to read," indicating persisting issues with the paper's clarity and flow.

**Reviewer Scores:**

NA

---

### Decision · Program_Chairs · 2026-01-26

Reject